# The Relationship Between Binge Eating Behavior and Psychological Pain in Patients with Major Depressive Disorder

**DOI:** 10.3390/bs15070842

**Published:** 2025-06-22

**Authors:** Aynur Özbay, Mehmet Emin Demirkol, Lut Tamam, Zeynep Namlı, Mahmut Onur Karaytuğ, Caner Yeşiloğlu

**Affiliations:** Department of Psychiatry, Faculty of Medicine, Çukurova University, Adana 01250, Türkiye; aynurozbay90@gmail.com (A.Ö.); medemirkol@cu.edu.tr (M.E.D.); ltamam@cu.edu.tr (L.T.); znamli@cu.edu.tr (Z.N.); mokaraytug@cu.edu.tr (M.O.K.)

**Keywords:** binge eating, depression, major depressive disorder, psychological pain, impulsivity

## Abstract

Major Depressive Disorder (MDD) is a chronic mental disorder characterized by anhedonia, loss of desire, guilt, suicidal thoughts, and appetite changes. It is reported that individuals with MDD resort to binge eating to escape from negative feelings. In this study, we aimed to determine the relationship between binge eating behavior and the concept of psychological pain associated with emotions such as shame, guilt, and anger in individuals with MDD. We conducted the study in the Psychiatry Outpatient Clinics of Balcalı Hospital, Çukurova University Faculty of Medicine. The sample consisted of 147 individuals with MDD without psychotic symptoms and 128 healthy controls with sociodemographic characteristics similar to the MDD group. We administered a sociodemographic data form, the Hamilton Depression Rating Scale (HDRS), Psychache Scale (PS), Tolerance for Mental Pain Scale-10 (TMPS-10), Barratt Impulsiveness Scale (BIS-11), and Eating Disorder Examination Questionnaire (EDE-Q-13). Eighty-two (55.7%) of the patients with MDD were diagnosed with binge eating disorder (BED). In the group of MDD patients with BED comorbidity, the EDE-Q-13 total, binging subscale, and HDRS scores were significantly higher than those of the other groups (*p* < 0.05 for each group), with large to very large effect sizes (e.g., EDE-Q-13 binging d = 1.04; HDRS d = 1.91; PS d = 1.22). There was no significant difference between the MDD groups (with and without BED) regarding the BIS and BIS subscales’ subscores, PS, and TMPS scores. For participants with MDD, there was a significant same-directional correlation between EDE-Q-13 binging, HDRS, BIS, and PS scores (*p* < 0.05 for each), with moderate to strong effect sizes (EDE-Q-13 binging and HDRS: r = 0.398, *p* < 0.001; binging and PS: r = 0.273, *p* < 0.001; binging and BIS: r = 0.233, *p* = 0.005; binging and TMPS-10: r = –0.257, *p* = 0.002). Additionally, a negative correlation was observed between TMPS and the scores for EDE-Q-13 binging, HDRS, BIS, and PS. A linear regression analysis indicated that depression severity and BMI were the strongest predictors of binge eating behavior (R^2^ = 0.243; f^2^ = 0.32). Based on our results, we concluded that the presence of binge eating behavior in patients with MDD is associated with more severe depressive symptoms, psychological pain, impulsivity, and lower tolerance to psychological pain. The finding that binge eating behavior was most strongly associated with depression severity and body mass index (BMI) supports the notion that binge eating behavior is a maladaptive attitude. Longitudinal studies comparing individuals with different BMIs in different clinical samples are needed to confirm our results.

## 1. Introduction

Major Depressive Disorder (MDD) is a chronic mental illness characterized by anhedonia, loss of interest, guilt, suicidal thoughts, and changes in appetite, sleep duration, and energy ([19]). MDD is a serious public health problem worldwide ([19]), causing high morbidity, mortality, and treatment costs when not treated appropriately ([11]). According to the World Health Organization’s Global Health Estimates on prevalent mental disorders, more than 300 million people worldwide have a diagnosis of MDD with an average prevalence of 4.2%, accounting for 10.3% of the total global burden of disease ([7]).

Appetite change, which is one of the neurovegetative symptoms of MDD, may occur as an increase in some patients and a decrease in others ([43]). Binge eating disorder (BED) is an eating disorder with a loss of control over eating behavior, recurrent episodes of binge eating, and the absence of compensatory behaviors ([3]; [8]). Although the etiology of BED is not yet fully known, it is accepted to develop due to the interaction of environmental and biological risk factors. Psychological factors such as family conflicts, high parental demands, life stressors, depressive symptoms, separation from a loved one, the loss of a friend or family member, the termination of a relationship, and sexual and physical abuse are important initiators for BED rather than appetite changes and hunger ([40]; [14]). In addition, difficulties in emotional control rather than cognition are at the forefront of the continuation of binge eating behavior ([25]).

[29] ([29]) reported the presence of at least one lifetime psychiatric comorbidity in 67% of individuals with BED and more than three in approximately 50%. Mood disorders and anxiety disorders are at the forefront among comorbid diagnoses ([26]). In addition, disorders characterized by poor impulse control, including attention deficit hyperactivity disorder (ADHD), personality disorders, and gambling disorder, frequently accompany BED. Individuals with BED display eating patterns characterized by a subjective loss of control over their eating behaviors. BED is often associated with increased reward sensitivity and impulsivity, which may lead to compulsive-like eating behavior ([8]). Many individuals affected by this disorder are overweight, as compensatory behaviors such as physical exercises and the use of laxatives do not occur. Some researchers explained this pattern of disinhibited eating by hypothesizing that BED represents a phenotype characterized by increased impulsivity within the obesity spectrum. [12] ([12]) stated that individuals with high reward sensitivity experience certain foods as very rewarding and have trouble resisting reward, resulting in a loss of control over eating. Apart from general impulsivity tendencies, [37] ([37]) found that high impulsivity, especially in motor and attention, is a risk factor for binge eating. Impulsivity, defined by different sub-dimensions, is an important factor in initiating and maintaining binge eating ([42]).

Psychological pain refers to the mental suffering that arises from experiences such as loss, exposure to traumatic events, disappointment, unexpected negative situations, and unmet basic needs ([22]). According to [35] ([35]), psychological pain is a persistent and distressing feeling that occurs when individuals negatively perceive their inability to protect themselves from shame or harm, fail to achieve something deemed vital, or experience a lack of love, support, or connection with others. Although there is increasing evidence supporting the idea that psychological pain represents an independent phenomenon with its neurobiological properties (altered activities in brain regions such as the thalamus, anterior and posterior cingulate cortex, the prefrontal cortex, cerebellum, and parahippocampal gyrus), there are also publications indicating that psychological pain is associated with various psychopathological conditions such as suicidal ideation, anxiety disorders, borderline personality disorder, and emotional dysregulation, primarily MDD ([49]; [35]; [48]; [24]; [27]). Although there are studies investigating the relationship of psychological pain with mental disorders such as obsessive–compulsive disorder (OCD), bipolar disorder, and MDD, to the best of our knowledge, there is not enough data on its relationship with eating disorders and eating attitudes ([17], [16]; [7]).

Eating disorders are not only characterized by impaired eating-related behaviors. Difficulties in tolerating negative emotional states, emotional dysregulation, and major depressive disorder often co-occur with eating disorders ([47]). Women with anorexia nervosa experience anguish and frustration due to dissatisfaction with their appearance, thus experiencing more intense depressive symptoms and psychological pain ([9]). Binge eating disorder is characterized by feelings of self-loathing, depression, or guilt after eating ([3]). [20] ([20]) reported that emotional states such as shame and distress are predictors of binge eating.

Negative emotions are involved in the onset and maintenance of binge eating behavior in BED, but it is not clear whether eating behavior reduces negative affects ([50]; [13]). Previous studies attributed participants’ binge eating behaviors to mood rather than hunger. ([38]) and [54] ([54]), who investigated a wide range of emotions in BED, reported that binge eating episodes were associated with feelings of anger, frustration, hurt, or loneliness. These concepts also overlap with Shneidman’s definition of psychological pain. Based on the hypothesis that binge eating behaviors may be a dysfunctional, emotion-focused coping strategy, we aimed to examine the relationship between psychological pain and binge eating behaviors in individuals with MDD. We hypothesized that binge eating behavior is exhibited more frequently and intensely in patients with MDD who experience more intense psychological pain. Our second hypothesis is that depression severity and binge eating behavior are correlated.

## 2. Materials and Methods

The sample consisted of 160 participants aged 18–65 who were diagnosed with MDD according to the Diagnostic and Statistical Manual of Mental Disorders (DSM-5) criteria and 131 healthy volunteers with similar sociodemographic characteristics to the patients who had no previous psychiatric diagnosis or treatment. Only literate participants were included for the application of self-report scales, all participants were informed of the study, and their written informed consent was obtained before participation. The Non-Interventional Clinical Research Ethics Committee of Çukurova University Faculty of Medicine approved the study (date 1 September 2023 and number 136). This research adhered to the ethical principles of the Helsinki Declaration ([53]) for human studies and was conducted between 1 October 2023 and 15 January 2024.

In the MDD group, those diagnosed with intellectual disability, neurocognitive disorders, comorbid mental disorders, eating disorders other than binge eating disorder, and those diagnosed with chronic physical diseases such as malignancy and endocrine diseases were excluded by examining hospital files, self-statements of the participants, and health system records because of their confounding effects on eating attitudes. After psychiatric interviews based on DSM-5 criteria, 10 individuals with comorbid anxiety disorders and 3 with alcohol use disorder were excluded. In addition, those diagnosed with atypical depression, one of the determinants of MDD, were also not included due to the presence of increased appetite among the diagnostic criteria. In the control group, participants with a diagnosis of mental disorder, psychotropic drug users, and those with a diagnosis of chronic physical illness were also excluded. After psychiatric interviews based on DSM-5 criteria, two participants with panic disorder and one with OCD were excluded. The final sample consisted of 147 participants with MDD and 128 healthy volunteers.

We performed an a priori power analysis to compare scale scores between the MDD and control groups. With a medium effect size (Cohen’s d = 0.50), a power of 0.95, and a margin of error of 0.05 (*p* = 0.05), the minimum sample size required in a single group was calculated as 105 for the patient and control groups, totaling 210. With 275 participants, we assumed that the sample had sufficient power.

### 2.1. Measures

The primary author, a final-year student (MD) specializing in mental health and diseases, conducted interviews with all participants lasting approximately 40 to 60 min, under the supervision of the secondary author, an associate professor in psychiatry. In the first stage, the author informed the participants of the study, obtained their written informed consent, and completed the sociodemographic data form and the HDRS during a face-to-face interview. In the second stage, self-report scales were administered to the participants.

Sociodemographic Data Form: This form consists of age, sex, marital status, place of residence, family type, occupational and educational status, income level, diagnosis of physical illness, family history of mental disorders, history of suicide attempts, smoking, alcohol and substance use, and height and weight measurements.

The first author took height and weight measurements after the psychiatric interview. The body mass index (BMI) was calculated by dividing weight in kilograms by the square of height in meters (kg/m^2^). The BMI categorization of the participants was based on the [52]’s ([52]) classification of underweight (BMI < 18.5 kg/m^2^), normal weight (BMI 18.5–24.9 kg/m^2^), overweight (BMI 25.0–29.9 kg/m^2^), and obese (BMI ≥ 30.0 kg/m^2^). The participants’ income levels were divided into below and above minimum wage based on the minimum wage determined by the Turkish government (approximately USD 550 per month). The family type was divided into two categories according to household structure. When the household consisted of parents and children, it was defined as a nuclear family; when it also consisted of other relatives, it was defined as an extended family. Place of residence was divided into two groups: “provincial center” when they were city centers and “settlement units smaller than the provincial center” when they were smaller in scale, such as districts, towns, or villages.

The Hamilton Depression Rating Scale (HDRS): The HDRS is a 17-item scale that determines the severity of depressive symptoms and is completed by the clinician. The first item of the scale assesses the depressed mood in a range (grief, hopelessness, helplessness, and worthlessness) from 0 (none) to 4 (patient expresses these symptoms verbally and behaviorally). The scale’s total score varies between 0 and 53, and items such as insight, weight loss, somatic symptoms, sexual desire, early morning awakening, midnight awakening, and difficulty falling asleep are given a score between 0 and 2 points, and other items are given a score between 0 and 4 points ([55]). In a Turkish validity and reliability study, [1] ([1]) found the inter-rater internal consistency coefficient to be in the range of 0.87–0.98.

The Psychache Scale (PS): The PS is a 13-question self-report scale that assesses the severity of psychological pain. The first item of the PS is “I feel psychological pain”. It is scored on a five-point Likert-type, and an increase in the total score indicates an increase in the severity of psychological pain ([28]). [15] ([15]) determined a Cronbach’s alpha value of 0.98 in the Turkish validity and reliability study.

The Tolerance for Mental Pain Scale-10 (TMPS-10): The TMPS-10 assesses the ability to tolerate psychological pain. The first item of the TMPS is “I believe that my pain will go away”. It is a 10-item self-report scale with a five-point Likert-type score. Higher scores correspond to a higher tolerance to psychological pain ([36]). In the Turkish validity and reliability study, [18] ([18]) determined the Cronbach’s alpha value to be 0.96.

The Eating Disorder Examination Questionnaire Short Form (EDE-Q-13): The EDE-Q-13 is a self-report scale consisting of 13 items and five subscales, including eating restraint (1, 2, and 3), shape and weight over-evaluation (4 and 5), body dissatisfaction (6 and 7), binging (8, 9, and 10) and purging (11, 12, and 13). Item 9 is “I have felt out of control when eating”. The EDE-Q-13 score is evaluated according to the total score of the items in the whole scale and each subscale. The subscale scores are divided by the number of items, and the total score is calculated by dividing the sum of all subscale scores by the number of subscales ([32]). Higher scores indicate higher levels of eating-related psychopathology. [21] ([21]) conducted a Turkish validity and reliability study and found a Cronbach’s alpha coefficient of 0.89.

The Barratt Impulsiveness Scale Short Form (BIS-11): The BIS-11 is a self-report scale comprising 15 items and three subscales: non-planning, motor impulsivity, and attention impulsivity. Attention impulsivity is making quick decisions, motor impulsivity is taking action without thinking, and non-planning is focusing on the moment or not thinking about the future. The first item of the BIS-11 is “I plan tasks carefully”. The higher the total score, the higher the level of impulsivity ([5]). [46] ([46]) conducted a Turkish validity and reliability study, and the Cronbach’s alpha value was 0.82.

### 2.2. Statistical Analysis

#### Procedure

This study was conducted with a cross-sectional questionnaire design. Data were collected in three groups: (1) individuals with MDD and BED (MDD+BED+), (2) individuals with MDD without BED (MDD-only), and (3) healthy controls.

Descriptive statistics were summarized as the mean ± standard deviation for continuous variables and as a number and percentage for categorical variables. The normality of the distribution of continuous variables was assessed using skewness and kurtosis values (acceptable range: −1.5 to +1.5) ([45]). The chi-square test was used for categorical variable comparisons between groups.

To test the hypothesis that binge eating behavior is more frequent and intense in individuals with MDD who experience more intense psychological pain (hypothesis 1), the EDE-Q-13 binging subscale, PS, and TMPS-10 scores were compared between the three groups with a one-way ANOVA. Homogeneity of variance was evaluated with Levene’s test; the Tukey test was used when homogeneity was achieved, and the Tamhane test was used for post hoc analysis when homogeneity was not achieved.

While assessing the hypothesis that depression severity and binge eating behavior (hypothesis 2) are correlated, a Pearson correlation analysis was performed between the scores of the EDE-Q-13 binging subscale and HDRS, PS, BIS-11, and TMPS-10 in the MDD group.

We performed a linear regression analysis to determine the factors predicting binge eating behavior in the MDD group (with and without BED). The dependent variable was the EDE-Q-13 binging subscale score. The independent variables were the HDRS, PS, and TMPS-10 scores and BMI. Variance Inflation Factor (VIF) values were examined for multicollinearity assessment; a VIF value below five was considered appropriate.

The statistical significance level was accepted as *p* < 0.05 in all analyses. Statistical analyses were performed using IBM SPSS 22.0 software (IBM Corporation, Armonk, NY, USA).

## 3. Results

The participants with MDD were divided into two groups: those with and without comorbid BED (82 and 65 participants). Table 1 compares the sociodemographic characteristics of the groups. There was no statistically significant difference between the groups regarding sex, marital status, employment status, place of residence, income level, and family type (*p* > 0.05 for each). These findings reveal that the groups are comparable in terms of demographic characteristics.

There was a statistically significant association between the groups and the BMI category (χ^2^ (6, *n* = 274) = 33.48, *p* < 0.001). A post hoc analysis revealed that individuals in the MDD+BED+ group were significantly more likely to have obesity compared to the MDD-only and control groups. In contrast, participants in the control group were significantly more likely to have a normal BMI.

Table 2 shows comparisons between the two groups: 147 participants with MDD and 128 healthy controls. There were statistically significant differences between the groups for EDE-Q-13 binging, HDRS, TMPS-10, PS, and BIS-11 subscales and total scores and BMI (*p* < 0.05 for each).

In addition to assessing statistical significance, the effect sizes were calculated for all comparisons. The difference in EDE-Q-13 binging scores between the MDD group and the control group resulted in a large effect size (Cohen’s d = 1.04). The HDRS and the PS scores demonstrated very large effect sizes (d = 1.91 and d = 1.22, respectively). Medium to large effect sizes were observed in impulsivity scores, including the BIS attention (d = 0.58) and BIS non-planning (d = 0.62) subscales. Additionally, the TMPS-10 score revealed a large negative effect (d = −1.23), indicating that the control participants reported higher scores than those in the MDD group.

Table 3 compares the PS, EDE-Q-13 total score, EDE-Q-13 binging subscale, TMPS-10, and HDRS scores of the participants (three groups). The EDE-Q-13 total and EDE-Q-13 binging scores of the MDD+BED+ group were significantly higher than the MDD-only and control groups (*p* < 0.05 for each comparison). The PS scores of the control group were significantly lower, and the TMPS-10 scores were significantly higher than those of the MDD+BED+ and MDD-only groups (*p* < 0.05 for each comparison). There was no significant difference between the MDD+BED+ and MDD-only groups regarding the TMPS-10 scores. There was a significant difference between all groups regarding the HDRS scores; the highest score was found in the MDD+BED+ group, and the lowest was in the control group (*p* < 0.05 for each).

Table 4 compares the BIS-11 total and subscale scores (attention, motor, and non-planning impulsivity) of the participants (three groups). The BIS attention and BIS non-planning scores were significantly higher in the MDD groups (MDD+BED+ and MDD-only groups) compared to the control group (*p* < 0.05), and the difference between the MDD+BED+ and the MDD-only groups was not statistically significant. The three groups had no significant differences regarding the BIS motor scores (*p* = 0.073). The BIS total score was significantly higher in the MDD+BED+ group than in the MDD-only and control groups; the MDD-only group also had significantly higher scores than the control group (*p* < 0.05).

ANOVA analyses revealed significant effect sizes. The difference in the EDE-Q-13 total scores between the groups showed a large effect (η^2^ = 0.36), indicating substantial differences among the three groups. Moderate to large effects were observed for the BIS non-planning and BIS total scores (η^2^ = 0.10 for each). The effect size for the BIS attention subscale was moderate (η^2^ = 0.08), while the BIS motor impulsivity subscale had a small effect (η^2^ = 0.02).

Table 5 shows the correlation analyses between the scale scores of the MDD groups (MDD+BED+ and MDD-only groups) (*n*:147). There was a significant positive correlation between the EDE-Q-13 binging and HDRS, BIS-11, and PS scores. There was also a significant negative correlation between the TMPS-10 and EDE-Q-13 binging, HDRS, BIS, and PS scores (*p* < 0.05).

The Pearson correlation coefficients, when interpreted as effect sizes, indicated moderate to strong associations among the variables. For example, binge eating showed a positive correlation with the HDRS at r = 0.398, with the PS at r = 0.273 and the BIS at r = 0.233. Conversely, it displayed a negative correlation with the TMPS-10 at r = −0.257. The strongest association was observed between the TMPS-10 and PS, which had a large negative effect size of r = −0.623.

Table 6 presents the linear regression model of the factors predicting binge eating behavior. Our regression model was statistically significant (F(4,142) = 11.407, *p* < 0.001). The model explained 24.3% of the variance in the EDE-Q-13 binging score (R^2^ = 0.243; adjusted R^2^ = 0.222). The HDRS total score and BMI were the strongest predictors of the EDE-Q-13 binging score (the dependent variable; *p* = 0.011, *p* < 0.001), respectively. The PS and TMPS-10 scores did not have a significant predictive role in the model [zn1]. No multicollinearity problem was observed (VIF < 2.5). According to Cohen’s f^2^ criteria, this corresponds to a medium to large effect size (f^2^ = 0.32).

## 4. Discussion

MDD is a common psychiatric disorder that causes impairment, reduces productivity, and threatens public health. The psychological pain that often accompanies MDD negatively affects the prognosis. Consistent with our hypotheses, we determined that psychological pain and depressive symptoms are more severe in individuals with MDD when the diagnosis of BED is added and that individuals with BED tolerate psychological pain worse. [33] ([33]) evaluated the frequency of emotional eating associated with feelings of anxiety, sadness, loneliness, fatigue, anger, and happiness in individuals with overweight (BMI ≥ 25) diagnosed with BED. They demonstrated a significant relationship between eating disorder symptoms and the severity of depressive symptoms ([33]). The risk of being diagnosed with BED is approximately five times higher in individuals with MDD than in those without MDD ([39]). [33] ([33]) did not determine the comorbid psychiatric disorders of the participants. Our findings indicate that binge eating behavior was significantly more pronounced in the MDD+BED+ and MDD-only groups compared to the healthy controls, aligning with previous research. Furthermore, our results show that the MDD+BED+ group experienced more severe depressive symptoms than the MDD-only group. This confirms that individuals with depression are more susceptible to emotional eating and are at a higher risk for developing eating disorders. These results support the existence of a bidirectional relationship between binge eating behaviors and depressive symptoms ([41]).

We also examined the factors associated with binge eating behavior and found that the strongest predictors were the severity of depressive symptoms and BMI. The prevalence of obesity in the group with MDD+BED+ was higher than in the MDD-only and control groups. For individuals with BED, weight gain is a common outcome as they typically do not engage in compensatory behaviors ([29]). Research indicates that BED is strongly linked to obesity, with individuals affected by BED often having a higher BMI compared to the general population ([29]; [26]). It has also been reported that individuals with a high BMI exhibit increased reward sensitivity, making them more likely to pursue food for its rewarding properties ([12]; [37]). This supports our finding that BMI is one of the strongest predictors of BED symptoms. In contrast, eating behavior tends to be more regular and is regulated by homeostatic mechanisms in individuals without BED ([30]). This explains why individuals in the control group had more normal BMI values compared to those in the MDD group. Moreover, negative affect is a precursor to disordered eating behaviors and an important vulnerability factor in the development of eating disorders. Binge eating is associated with both inadequate emotion regulation and expectations that eating will alleviate negative emotions. The relief of the negative affect that increases before the eating behavior after the binge behavior may lead to binge eating serving as a method of coping with negative affect and reinforcing this behavior ([6]; [51]). The efficacy of negative emotions, maladaptive coping skills, and negative reinforcement in instigating and perpetuating binge eating may elucidate the significant association between binge eating behavior and depressive symptom severity in our study.

The relationship between the affect regulation model described above and binge eating is well documented in the literature ([6]; [2]). Consistent with the affect regulation model, [6] ([6]) suggested that binge eating reduces negative affect and guilt, albeit in the short term ([6]). To the best of our knowledge, the relationship between the concept of psychological pain, which is closely related to negative affect, and BED has only been examined by [47] ([47]). [47] ([47]) compared 74 people with eating disorders and 90 healthy individuals regarding the intensity of psychological pain and found that psychological pain was more severe in those with eating disorders than in those without eating disorders. Among participants with eating disorders, those who experienced more intense psychological pain had more severe depressive symptoms, increased eating behavior problems, and more severe suicidal thoughts. Similarly to [47] ([47]), binge eating behavior, depressive symptom severity, and psychological pain were correlated in patients with MDD in our study. However, when patients with MDD were classified according to whether BED accompanied them, the severity of psychological pain was similar, while the MDD+BED+ group had more severe depressive symptoms. Our results suggest that the relationship between negative affect and binge eating is not specific to individuals with eating disorders at the clinical level, as emphasized by [6] ([6]). Therefore, questioning binge eating behavior in the treatment process of patients with MDD, even if an eating disorder does not accompany them, may help to identify maladaptive behaviors.

Although individuals turn to binge eating behavior to cope with negative emotions, there are controversial findings on whether this behavior actually reduces emotional distress—with some evidence suggesting only temporary relief followed by increased negative affect. [44] ([44]) evaluated 33 women classified as overweight with BED according to the DSM-IV criteria and examined possible associations between binge eating behavior and pleasantness, mood, guilt, and arousal states while the participants were binge eating or eating regularly. [44] ([44]) concluded that negative mood was significantly stronger after binge eating than after regular eating. [13] ([13]) compared the displeasure before, during, and after binge eating periods and regular eating periods at 2 min intervals and reported that BED caused short-term positive feelings, and this reinforced binge eating behavior. Our study revealed that participants with MDD were less able to tolerate psychological pain than the healthy controls. We also found that the presence of BED among participants with MDD was associated with less tolerance to psychological pain, and tolerance to psychological pain was lower in patients with severe depressive symptoms. The lack of a significant predictor effect of psychological pain on binge eating behavior in our depressive patients supports the hypothesis that eating for emotion regulation is a maladaptive coping skill and that the use of non-result-oriented coping skills does not improve psychiatric complaints ([34]).

Impulsivity is a multidimensional symptom (rapid decision-making, lack of control, inability to make decisions in advance, inability to postpone, and thrill-seeking) that plays a role in the emergence and eroticization of eating disorders and is also associated with binge eating symptoms ([10]; [31]). BED has been increasingly recognized as a multifactorial disorder involving emotional dysregulation, impulsivity, and impaired control over eating behaviors ([8]). It is noteworthy that individuals with BED have more reward sensitivity and reactive behaviors than those without BED. [23] ([23]) reported that impulsivity moderated the relationship between negative affect and increased calorie intake. Our results determine that the severity of impulsivity is not significantly different between those with MDD and BED together and those with MDD alone; this finding supports the hypothesis that BED is not in the same cluster with impulsivity and related disorders and that binge eating behavior serves the purpose of emotion regulation rather than impulsivity in classification systems such as the DSM-5 ([44]; [3]). In addition, the concepts of psychological pain and tolerance to psychological pain are closely related to MDD and negative affect. However, impulsivity is a more independent concept classified as a state or trait. In our study, the questioning of general impulsivity tendencies instead of state impulsivity measurement may have reflected the personality characteristics of participants with MDD independently of a BED diagnosis ([4]).

Our study has some limitations. First, the depressive group consisted of individuals with MDD who were admitted to a psychiatry outpatient clinic of a university hospital. Second, the depressive group does not represent cases of mild depressive disorder. Participants with MDD who were admitted to the emergency department or who underwent inpatient treatment for suicide attempts were also excluded from the sample. Third, since our study has a cross-sectional design, it is not possible to determine a cause-and-effect relationship. Finally, although MDD is more common in women than men, the diagnosis of MDD was not evaluated according to sex in our study. Future studies can be designed to reveal differences between the sexes.

Our study presents a transdiagnostic approach to MDD and BED comorbidity by investigating the relationship between psychological pain and impulsivity. In addition, we evaluated the relationship between BMI, depressive symptoms, psychological pain, and psychological pain tolerance on BED with a multifactorial approach. The clinical contributions of our study include demonstrating that eating disorders should be routinely questioned in the evaluation of patients with MDD and the severity of psychological pain should be taken into account in clinical follow-up. Our study also suggests that approaches to improving emotion regulation skills in MDD and BED comorbidity may be useful.

## 5. Conclusions

Our results demonstrate that MDD can be characterized by a wide range of symptoms, such as severe psychological pain, impulsivity, and binge eating behavior, in addition to typical depressive symptoms. The difference in depressive symptom severity in patients with MDD+BED+ compared to the MDD-only group suggests that negative affect plays a leading role and that individuals resort to binge eating for affect regulation. The finding that binge eating behavior is not associated with the trait of impulsivity may be interpreted as a maladaptive coping strategy instead of losing control over eating. Especially in patients with MDD with high BMI, binge eating behavior should be questioned, and adaptive affect regulation methods should be targeted in treatment. Future studies could examine these findings using different samples and longitudinal designs.

## Figures and Tables

**Table 1 behavsci-15-00842-t001:** Sociodemographic characteristics of participants.

Variables	MDD+BED+ (*n* = 82)	MDD-Only (*n* = 65)	Control (*n* = 128)	Effect Size (Cramér’s)	*p*
**Sex *n* (%)**					
Female	60 (73.2)	42 (64.6)	91 (71.1)		0.50
Male	22 (26.8)	23 (35.4)	37 (28.9)		
**Marital Status *n* (%)**					
Married	41 (50.0)	34 (52.3)	54 (42.2)		0.32
Single	41 (50.0)	31 (47.7)	74 (57.8)		
BMI	28.25 ± 5.64	24.95 ± 4.88	24.06 ± 4.14		0.36
**BMI Categorizations *n* (%)**					
Underweight	0 (0)	5 (7.8)	7 (5.5)	0.247	*p* < 0.001
Normal Weight	25 (30.5)	31 (48.4)	77 (60.2)		
Overweight	32 (39)	19 (29.7)	34 (26.6)		
Obesity	25 (30.5)	9 (14.1)	10 (7.8)		
**Employment Status *n* (%)**					
Employed	43 (52.4)	30 (46.2)	71 (55.5)		0.47
Unemployed	39 (47.6)	35 (53.8)	57 (44.5)		
**Place of Residence *n* (%)**					
Provincial Center	65 (79.3)	53 (81.5)	102 (79.7)		0.93
Smaller than Provincial Center	17 (20.7)	12 (18.5)	26 (20.3)		
**Income Level *n* (%)**					
Below Monthly Minimum Wage	27 (32.9)	17 (26.2)	46 (35.9)		0.39
Above Monthly Minimum Wage	55 (67.1)	48 (73.8)	82 (64.1)		
**Family Type *n* (%)**					
Immediate Family	71 (86.6)	54 (83.1)	117 (91.4)		0.21
Extended Family	11 (13.4)	11 (16.9)	11 (8.6)		

Wage refers to the legally mandated lowest salary that an employer is required to pay employees in the country. BMI: Body Mass Index, MDD: Major Depressive Disorder, BED: Binge Eating Disorder.

**Table 2 behavsci-15-00842-t002:** Comparison of scale scores between participants with MDD and control group.

Mean (SD)	MDD (Mean ± SD)	Control (Mean ± SD)	F	t	df	Effect Size (d)	*p*
**EDE-Q-13 binging**	2.20 (1.97)	0.60 (0.80)	122.111	8.541	273	1.04	<0.001
**BIS—attention**	17.34 (4.11)	15.1 (3.58)	2.935	4.680	273	0.58	<0.001
**BIS—motor**	19.92 (4.69)	18.76 (4.03)	1.903	2.180	273		0.030
**BIS—non-planning**	26.51 (4.91)	23.6 (4.59)	1.007	5.031	273	0.62	<0.001
**BIS—total**	63.78 (10.61)	57.53 (9.78)	0.722	5.051	273	0.61	<0.001
**HDRS**	26.37 (8.16)	4.41 (2.84)	117.176	28.940	273	1.91	<0.001
**PS**	37.9 (13.49)	13.49 (1.11)	47.904	10.087	273	1.22	<0.001
**BMI**	26.79 (5.55)	24.06 (4.14)	7.852	4.566	273	0.55	<0.001
**TMPS-10**	32.12 (8.81)	41.90 (6.88)	7.714	−10.149	273	−1.23	<0.001

MDD: Major Depressive Disorder, EDE-Q-13: Eating Disorder Examination Questionnaire, BIS: Barratt Impulsiveness Scale, HDRS: Hamilton Depression Rating Scale, TMPS: Tolerance for Mental Pain Scale, PS: Psychache Scale, BMI: Body Mass Index, SD: Standard Deviation.

**Table 3 behavsci-15-00842-t003:** A comparison of the scale scores of the participants.

Variables	Group	Mean ± (SD)	Effect Size (η^2^)	Significant	Post Hoc
**PS**	^1^MDD+BED+	40.1 (12.9)	0.28	<0.001	**1>3 *** **2>3 *** 1>2
	^2^MDD-only	35.3 (13.7)			
	^3^Control	24.1 (8.1)			
**EDE-Q-13—Total**	^1^MDD+BED+	2.8 (0.9)	0.36	<0.001	**1>3 ^#^** **1>2 ^#^** 3>2
	^2^MDD-only	1.3 (0.8)			
	^3^Control	1.4 (0.8)			
**EDE-Q-13 Binging**	^1^MDD+BED+	3.3 (1.7)	0.50	<0.001	**1>3 *** **1>2 *** 2>3
	^2^MDD-only	0.7 (1.1)			
	^3^Control	0.6 (0.8)			
**TMPS-10**	^1^MDD+BED+	31.2 (8.6)	0.28	<0.001	**3>1 ***
	^2^MDD-only	33.2 (8.9)			2>1
	^3^Control	41.9 (6.8)			**3>2 ***
**HDRS**	^1^MDD+BED+	28.1 (7.8)	0.76	<0.001	**1>3 *** **1>2 *** **2>3 ***
	^2^MDD-only	24.1 (8)			
	^3^Control	4.4 (2.8)			

Note: PS: Psychache Scale, EDE-Q-13: Eating Disorder Examination Questionnaire, HDRS: Hamilton Depression Rating Scale, TMPS: Tolerance for Mental Pain Scale, sd: Standard Deviation, MDD: Major Depressive Disorder, BED: Binge Eating Disorder. ^#^ Tukey’s test; * Tamhane test. Statistically significant differences are highlighted in bold.

**Table 4 behavsci-15-00842-t004:** Comparison of impulsivity dimensions of participants.

Variables	Group	Mean ± SD	Effect Size (η^2^)	Significant	Post Hoc
**BIS—attention impulsivity**	^1^MDD+BED+	17.6 (3.8)	**0.08**	**<0.001**	**1>3 ^#^** 1>2 **2>3 ^#^**
	^2^MDD-only	16.9 (4.3)			
	^3^Control	15.1 (3.5)			
**BIS—motor impulsivity**	^1^MDD+BED+	20.3 (4.8)		0.073	1>3 1>2 2>3
	^2^MDD-only	19.4 (4.5)			
	^3^Control	18.7 (4.0)			
**BIS—non-planning impulsivity**	^1^MDD+BED+	27.1 (5.0)	**0.10**	**<0.001**	**1>3 *** 1>2 **2>3 ***
	^2^MDD-only	25.6 (4.6)			
	^3^Control	23.6 (4.5)			
**BIS—total**	^1^MDD+BED+	65.1 (10.4)	**0.10**	**<0.001**	**1>3 ***
	^2^MDD-only	62.0 (10.6)			1>2
	^3^Control	57.5 (9.7)			**2>3 ***

Note: BIS: Barratt Impulsiveness Scale, sd: Standard Deviation, MDD: Major Depressive Disorder, BED: Binge Eating Disorder. ^#^ Tukey’s test; * Tamhane test. Statistically significant differences are highlighted in bold.

**Table 5 behavsci-15-00842-t005:** Correlation of scale scores in group diagnosed with Major Depressive Disorder (*n* = 147).

	EDE-Q-13 Binging	TMPS-10	HDRS	PS
**TMPS-10**				
r	−0.257 **			
p	0.002			
**HDRS**				
r	0.398 **	−0.411 **		
p	<0.001	<0.001		
**PS**				
r	0.273 **	−0.623 **	0.625 **	
p	0.001	<0.001	<0.001	
**BIS**				
r	0.233 **	−0.283 **	0.283 **	0.308 **
p	0.005	0.001	0.001	<0.001

Note: HDRS: Hamilton Depression Rating Scale, TMPS: Tolerance for Mental Pain Scale, EDE-Q-13: Eating Disorder Examination Questionnaire, PS: Psychache Scale, BIS: Barratt Impulsiveness Scale. ** *p* < 0.001.

**Table 6 behavsci-15-00842-t006:** Linear regression analysis of factors predicting binge eating behavior in participants diagnosed with MDD.

Model	Coefficients		Sig.	95.0% Confidence Interval for B		VIF
	B	Standard Error		Lower Bound	Upper Bound	
**Constant**	−0.541	1.260	0.436	−3.032	1.950	
**PS**	−0.002	0.016	0.890	−0.034	0.030	2.241
**HDRS**	0.051	0.024	**0.011**	0.003	0.099	1.819
**BMI**	0.124	0.030	<**0.001**	0.064	0.183	1.125
**TMPS-10**	−0.033	0.021	0.103	−0.074	0.009	1.646

Note: HDRS: Hamilton Depression Rating Scale, TMPS: Tolerance for Mental Pain Scale, PS: Psychache Scale, BMI: Body Mass Index, MDD: Major Depressive Disorder. Statistically significant differences are highlighted in bold.

## Data Availability

The data supporting the findings of this study are not publicly available due to privacy and ethical restrictions but are securely retained by the authors. The data may be made available by the corresponding author upon reasonable request and with appropriate ethical approval.

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
