# Peer review of "The Relationship Between Binge Eating Behavior and Psychological Pain in Patients with Major Depressive Disorder"

_behavsci, 2025, doi:10.3390/bs15070842_

Round 1
Reviewer 1 Report
Comments and Suggestions for Authors
Dear Authors,
Considering the huge diffusion of Semaglutide and other glp1RAs, the manuscript is very interesting . However, it should be improved in order to be published.
Firstly, in the abstract a conclusion is needed. Follow the guidelines of the journal.
Secondly, with regard to the manuscript, all references should be updated, up to the last ten years maximum. The Introduction should be better organised, I would suggest a division in paragraphs. The same for the Discussion. From a practical point of view, what do Authors suggest for the clinical management of patients? Maybe could be interested to hypothesize a treatment panel and gite guidelines if there are.
Figures should be inserted and Tables improved.
Among data available, could you specify if there were patient with a diagnosis of SUD? Which type? Might these be a risk factors? Or a prognostic element?
Please add the following refs in the Discussion:
Exploring the association between suicidal thoughts, self-injury, and GLP-1 receptor agonists in weight loss treatments: Insights from pharmacovigilance measures and unmasking analysis.Guirguis A, Chiappini S, Papanti P GD, Vickers-Smith R, Harris D, Corkery JM, Arillotta D, Floresta G, Martinotti G, Schifano F. Eur Neuropsychopharmacol. 2024 May;82:82-91. doi: 10.1016/j.euroneuro.2024.02.003. Epub 2024 Mar 19.
Exploring the nexus of binge eating disorder (BED), New Psychoactive Substances (NPS), and misuse of pharmaceuticals: charting a path forward. Chiappini S, Papanti Pelletier GD, Vickers-Smith R, Corkery JM, Guirguis A, Martinotti G, Schifano F.Expert Opin Pharmacother. 2023 Sep-Dec;24(18):1915-1918. doi: 10.1080/14656566.2023.2271389. Epub 2024 Jan 5.
Is There a Risk for Semaglutide Misuse? Focus on the Food and Drug Administration's FDA Adverse Events Reporting System (FAERS) Pharmacovigilance dataset. Chiappini S, Vickers-Smith R, Harris D, Papanti Pelletier GD, Corkery JM, Guirguis A, Martinotti G, Sensi SL, Schifano F. Pharmaceuticals (Basel). 2023 Jul 11;16(7):994. doi: 10.3390/ph16070994.
Author Response
Comment 1:
Firstly, in the abstract a conclusion is needed. Follow the guidelines of the journal.
Response 1:
To address this comment, we have added a concluding statement to the abstract, as per journal requirements. Please see the related sentences below.
Based on our results, we concluded that the presence of binge eating behavior in patients with MDD is associated with more severe depressive symptoms, psychological pain, impulsivity, and lower tolerance to psychological pain. The finding that binge eating behavior was most strongly associated with depression severity and body mass index (BMI) supports that binge eating behavior is a maladaptive attitude. Longitudinal studies comparing individuals with different BMIs in different clinical samples are needed to confirm our results.
Comment 2:
Secondly, with regard to the manuscript, all references should be updated, up to the last ten years maximum.
Response 2:
We added updated references to our article. Please find some of the new references below:
- Ciwoniuk, N., Wayda-Zalewska, M., & Kucharska, K. (2023). Distorted body image and mental pain in anorexia nervosa. International Journal of Environmental Research and Public Health, 20, 718.
- Chiappini, S., Papanti Pelletier, G. D., Vickers-Smith, R., Corkery, J. M., Guirguis, A., Martinotti, G., & Schifano, F. (2023).
- Fang, S., Zhong, R., Zhou, S., Xu, J., Liu, Q., Wu, W., Li, H., & Wang, X. (2025). Multiple pathways to suicide: A network analysis based on three components of psychological pain. Journal of Affective Disorders, 372, 77–85.
- Wonderlich, J. A., Crosby, R. D., Engel, S. G., Crow, S. J., Peterson, C. B., Le Grange, D., & Fischer, S. (2022). Negative affect and binge eating: Assessing the unique trajectories of negative affect before and after binge‐eating episodes across eating disorder diagnostic classifications. International Journal of Eating Disorders, 55(2), 223–230.
Comment 3:
The Introduction should be better organised, I would suggest a division in paragraphs. The same for the Discussion.
Response 3:
To address this comment, we reorganized both the introduction and discussion. Please find some of the introduction’ new and reorganized paragraphs below:
Psychological pain refers to the mental suffering that arises from experiences such as loss, exposure to traumatic events, disappointment, unexpected negative situations, and unmet basic needs (Fang et al., 2025). According to Meerwijk et al. (2013), psychological pain is a persistent and distressing feeling that occurs when individuals negatively perceive their inability to protect themselves from shame or harm, fail to achieve something deemed vital, or experience a lack of love, support, or connection with others. Although there is increasing evidence supporting the idea that psychological pain represents an independent phenomenon with its neurobiological properties (altered activities in brain regions such as the thalamus, anterior and posterior cingulate cortex, the prefrontal cortex, cerebellum, and parahippocampal gyrus), there are also publications indicating that psychological pain is associated with various psychopathological conditions such as suicidal ideation, anxiety disorders, borderline personality disorder, and emotional dysregulation, primarily MDD (Heeringen et al., 2010; Meerwijk et al. 2013; Tossani 2013; Fertuck et al., 2016; Guidi et al., 2019). Although there are studies investigating the relationship of psychological pain with mental disorders such as obsessive-compulsive disorder (OCD), bipolar disorder, and MDD, to the best of our knowledge, there is not enough data on its relationship with eating disorders and eating attitudes (Demirkol et al. 2019; Demirkol et al., 2020; Chen et al., 2023).
Eating disorders are not only characterized by impaired eating-related behaviors. Difficulties in tolerating negative emotional states, emotional dysregulation, and major depressive disorder often co-occur with eating disorders (Tomba et al., 2021). Women with anorexia nervosa experience anguish and frustration due to dissatisfaction with their appearance, thus experiencing more intense depressive symptoms and psychological pain (Ciwoniuk et al., 2023). Binge eating disorder is characterized by feelings of self-loathing, depression, or guilt after eating (APA 2013). Duarte et al. (2017) reported that emotional states such as shame and distress are predictors of binge eating.
Some of the reorganized paragraphs of the discussion are:
We also examined the factors associated with binge eating behavior and found that the strongest predictors were the severity of depressive symptoms and BMI. The prevalence of obesity in the group with both MDD+ BED+ was higher than in the MDD-only and control groups. For individuals with BED, weight gain is a common outcome, as they typically do not engage in compensatory behaviors (Hudson et al., 2017). Research indicates that BED is strongly linked to obesity, with individuals affected by BED often having a higher BMI compared to the general population (Hudson et al., 2017; Grilo et al., 2021). It has also been reported that individuals with a high BMI exhibit increased reward sensitivity, making them more likely to pursue food for its rewarding properties (Dawe and Loxton, 2004; Meule and Platte, 2023). This supports our finding that BMI is one of the strongest predictors of BED symptoms. In contrast, eating behavior tends to be more regular and is regulated by homeostatic mechanisms in individuals without BED (Kessler et al., 2013). This explains why individuals in the control group had more normal BMI values compared to those in the MDD group. Moreover, negative affect is a precursor to disordered eating behaviors and an important vulnerability factor in the development of eating disorders. Binge eating is associated with both inadequate emotion regulation and expectations that eating will alleviate negative emotions. The relief of negative affect that increases before the eating behavior after the binge behavior may lead to binge eating serving as a method of coping with negative affect and reinforcing this behavior (Berg et al.2015; Wonderlich et al., 2021). These efficacy of negative emotions, maladaptive coping skills, and negative reinforcement in instigating and perpetuating binge eating may elucidate the significant association between binge eating behavior and depressive symptom severity in our study.
The relationship between the affect regulation model described above and binge eating has been well documented in the literature (Berg et al., 2015; Ambwani et al., 2015). Consistent with the affect regulation model, Berg et al. (2015) suggested that binge eating reduces negative affect and guilt, albeit short-term (Berg et al., 2015). To the best of our knowledge, the relationship between the concept of psychological pain, which is closely related to negative affect, and BED has only been examined by Tomba et al. (2021). Tomba et al. (2021) compared 74 people with eating disorders and 90 healthy individuals on the intensity of psychological pain and found that psychological pain was more severe in those with eating disorders than in those without eating disorders. Among participants with eating disorders, those who experienced more intense psychological pain had more severe depressive symptoms, increased eating behavior problems, and more severe suicidal thoughts. Similar to Tomba et al. (2021), binge eating behavior, depressive symptom severity, and psychological pain were correlated in patients with MDD in our study. However, when patients with MDD were classified according to whether BED accompanied them, the severity of psychological pain was similar, while the MDD+ BED+ group had more severe depressive symptoms. Our results suggest that the relationship between negative affect and binge eating is not specific to individuals with eating disorders at the clinical level, as emphasized by Berg et al. (2015). Therefore, questioning binge eating behavior in the treatment process of patients with MDD, even if an eating disorder does not accompany them, may help to identify maladaptive behaviors.
Comment 4:
From a practical point of view, what do Authors suggest for the clinical management of patients?
Response 4:
To address this comment, we tried to explain the study's clinical implications and add suggestions for treatment processes. Please find the related sentences in the last paragraph of the discussion and the first paragraph of the conclusion.
Our study presented a transdiagnostic approach to MDD and BED comorbidity by investigating the relationship between psychological pain and impulsivity. In addition, we evaluated the relationship between BMI, depressive symptoms, psychological pain, and psychological pain tolerance on BED with a multifactorial approach. The clinical contributions of our study include demonstrating that eating disorders should be routinely questioned in the evaluation of patients with MDD and the severity of psychological pain should be taken into account in clinical follow-up. Our study also suggests that approaches to improve emotion regulation skills in the MDD and BED comorbidity may be useful.
Conclusions
Our results demonstrate that MDD can be characterized by a wide range of symptoms, such as severe psychological pain, impulsivity, and binge eating behavior, in addition to typical depressive symptoms. The difference in depressive symptom severity in patients with MDD+BED+ compared to the MDD-only group suggests that negative affect would play a leading role and that individuals resort to binge eating for affect regulation. The finding that binge eating behavior is not associated with the trait impulsivity may be interpreted as a maladaptive coping strategy instead of losing control over eating. Especially in patients with MDD with high BMI, binge eating behavior should be questioned, and adaptive affect regulation methods should be targeted in treatment. Future studies could examine these findings using different samples and longitudinal designs.
Comment 5:
Figures should be inserted and Tables improved.
Response 5:
We tried to improve the tables to address this comment. Please find the new versions of tables 1 and 2 below. We do not have any figures in the current study.
Table 1. Sociodemographic Characteristics of the Participants
|
Variables |
MDD+BED+ (n=82)
|
MDD only (n=65)
|
Control (n=128)
|
p |
|
Sex n(%) |
||||
|
Female |
60(73.2) |
42(64.6) |
91(71.1) |
0.50 |
|
Male |
22(26.8) |
23(35.4) |
37(28.9) |
|
|
Marital Status n(%) |
||||
|
Married |
41(50.0) |
34(52.3) |
54(42.2) |
0.32 |
|
Single |
41(50.0) |
31(47.7) |
74(57.8) |
|
|
BMI |
28.25±5.64 |
24.95±4.88 |
24.06±4.14 |
0.36 |
|
BMI Categorizations n(%) Underweight Normal weight Overweight Obese |
0(0) 25(30.5) 32(39) 25(30.5) |
5(7.8) 31(48.4) 19(29.7) 9(14.1) |
7(5.5) 77(60.2) 34(26.6) 10(7.8) |
p<0.001 |
|
Employment Status n(%) |
||||
|
Employed |
43(52.4) |
30(46.2) |
71(55.5) |
0.47 |
|
Unemployed |
39(47.6) |
35(53.8) |
57(44.5) |
|
|
Place of Residence n(%) |
||||
|
Provincial Center |
65(79.3) |
53(81.5) |
102(79.7) |
0.93 |
|
Smaller than Provincial Center |
17(20.7) |
12(18.5) |
26(20.3) |
|
|
Income Level n(%) |
||||
|
Below Monthly Minimum Wage |
27(32.9) |
17(26.2) |
46(35.9) |
0.39 |
|
Above Monthly Minimum Wage |
55(67.1) |
48(73.8) |
82(64.1) |
|
|
Family Type n(%) |
||||
|
Immediate Family |
71(86.6) |
54(83.1) |
117(91.4) |
0.21 |
|
Extended Family |
11(13.4) |
11(16.9) |
11(8.6) |
|
The wage refers to the legally mandated lowest salary that an employer is required to pay employees in the country.
BMI: Body Mass Index, MDD: Major Depressive Disorder, BED: Binge Eating Disorder
Table 2. Comparison of Scale Scores Between Participants with MDD and the Control Group
|
Mean (SD)
|
MDD
|
Control |
F |
t |
df |
p |
|
EDE-Q-13 Bingeing |
2.20 (1.97) |
0.60 (0.80) |
122.111 |
8.541 |
273 |
<0.001 |
|
BIS-attetion |
17.34 (4.11) |
15.1 (3.58) |
2.935 |
4.680 |
273 |
<0.001 |
|
BIS-motor |
19.92 (4.69) |
18.76 (4.03) |
1.903 |
2.180 |
273 |
0.030 |
|
BIS-nonplanning |
26.51 (4.91) |
23.6 (4.59) |
1.007 |
5.031 |
273 |
<0.001 |
|
BIS-total |
63.78 (10.61) |
57.53 (9.78) |
.722 |
5.051 |
273 |
<0.001 |
|
HDRS |
26.37 (8.16) |
4.41 (2.84) |
117.176 |
28.940 |
273 |
<0.001 |
|
PS |
37.9 (13.49) |
13.49 (1.11) |
47.904 |
10.087 |
273 |
<0.001 |
|
BMI |
26.79 (5.55) |
24.06 (4.14) |
7.852 |
4.566 |
273 |
<0.001 |
|
TMPS-10 |
32.12 (8.81) |
41.90 (6.88) |
7.714 |
-10.149 |
273 |
<0.001 |
MDD: Major Depressive Disorder, EDE-Q-13: Eating Disorder Examination Questionnaire, BIS: Barratt Impulsiveness Scale, HDRS: Hamilton Depression Rating Scale, TMPS: Tolerance for Mental Pain Scale, PS: Psychache Scale, BMI: Body Mass Index sd: standart deviation
Comment 6:
Among data available, could you specify if there were patient with a diagnosis of SUD? Which type? Might these be a risk factors? Or a prognostic element?
Response 6:
Patients with substance use disorder (SUD) and other mental disorders were not included in the study. We tried to explain the characteristics of the sample in the first two paragraphs of the materials and methods section.
The sample consisted of 160 participants aged 18-65 who were diagnosed with MDD according to the Diagnostic and Statistical Manual of Mental Disorders (DSM-5) criteria and 131 healthy volunteers with similar sociodemographic characteristics to the patients who had no previous psychiatric diagnosis or treatment.
In the MDD group, those diagnosed with intellectual disability, neurocognitive disorders, comorbid mental disorders, eating disorders other than binge eating disorder, and those diagnosed with chronic physical diseases such as malignancy and endocrine diseases were excluded by examining hospital files, self-statements of the participants and health system records because of their confounding effects on eating attitudes.
Comment 7:
Please add the following refs in the Discussion...
(Guirguis et al., 2024; Chiappini et al., 2023; Chiappini et al., 2023)
Response 7:
To address this comment, we tried to mention the related articles in the introduction and discussion. Please find the related sentences below:
Appetite change, which is one of the neurovegetative symptoms of MDD, may occur as an increase in some patients and a decrease in others (Stahl 2015). Binge eating disorder (BED) is an eating disorder with a loss of control over eating behavior, recurrent episodes of binge eating, and the absence of compensatory behaviors (APA 2013; Chiappini et al. 2023).
Individuals with BED display eating patterns characterized by a subjective loss of control over their eating behaviors. BED is often associated with increased reward sensitivity and impulsivity, which may lead to compulsive-like eating behavior (Chiappini et al., 2023).
BED has been increasingly recognized as a multifactorial disorder involving emotional dysregulation, impulsivity, and impaired control over eating behaviors (Chiappini et al., 2023).

Reviewer 2 Report
Comments and Suggestions for Authors
Abstract
Line 25 – “Additionally, a negative correlation was observed between TMPS and the scores of 25 other scales.” Can the authors please clarify whether this means all other scales? It may be helpful to make this more explicit for the reader.
Line 26 – “BED is a common comorbid eating disorder with MDD” this phrase can be deleted for conciseness.
Line 29 – “The fact that”. Please could the authors maintain scientific accuracy and tone here. I would advise this is revised to “Our results suggest” or “The finding that”…
Keywords
It would be beneficial to include ‘major depressive disorder’ here.
Introduction
Spelling error – line 27 “accptd”
Lines 60-61: “Individuals with BED have an eating pattern in which they experience a subjective loss of control over their eating behaviors, and many are overweight, as compensatory behaviors do not accompany the picture.” – please can the authors provide an example of compensatory behaviours in parentheses. Additionally, ‘do not accompany the picture’ is rather informal. I’d suggest this is revised accordingly.
The description offered with respect to psychological pain requires more detail. Firstly, the authors should attempt to provide a definition based on previous research. For example, Meerwijk et al. (2013) describe psychological pain as a lasting, unsustainable and unpleasant feeling resulting from negative appraisal of an inability to protect the self from shame or harm, an ability to achieve something that is considered of vital importance, or a deficiency in love, support or affiliation.
There is evidence to suggest that the posterior cingulate cortex plays a more prominent role in psychological (vs. physical) pain. See the below which may be helpful:
Meerwijk, E. L., Ford, J. M., & Weiss, S. J. (2013). Brain regions associated with psychological pain: implications for a neural network and its relationship to physical pain. Brain imaging and behavior, 7, 1-14.
Van Heeringen, K., Van den Abbeele, D., Vervaet, M., Soenen, L., & Audenaert, K. (2010). The functional neuroanatomy of mental pain in depression. Psychiatry Research: Neuroimaging, 181(2), 141-144.
I would recommend a paragraph being added which highlights the potential links between binge eating behaviour, psychological pain and major depressive disorder. There is evidence in this area. While ‘psychological pain’ is not always explicitly measured in such studies, this can be indirectly indicated via the intense emotions which encapsulate psychological pain:
Tomba, E., Tecuta, L., Gardini, V., & Lo Dato, E. (2021). Mental pain in eating disorders: An Exploratory controlled study. Journal of Clinical Medicine, 10(16), 3584.
Ciwoniuk, N., Wayda-Zalewska, M., & Kucharska, K. (2022). Distorted body image and mental pain in anorexia nervosa. International journal of environmental research and public health, 20(1), 718.
Duarte, C., Pinto‐Gouveia, J., & Ferreira, C. (2017). Ashamed and fused with body image and eating: Binge eating as an avoidance strategy. Clinical psychology & psychotherapy, 24(1), 195-202.
The authors have not included any predictions based on previous research – based on the literature available what do the authors speculate may be the relationship between psychological pain and binge eating behaviours in individuals with MDD?
Methodology
Please change the term “mental retardation” to “intellectual disability” or similar. This is an outdated term.
What design was factored into the G*power analysis? I presume a between-subjects design.
The participant sample numbers can be clearer to ease cognitive load on the reader. It would be best to state what the original sample was, then mention the participants excluded and that the “final sample consisted of XXX”.
Can you please provide an example item from the HDRS, PS, TMPS-10, EDE-Q-13 and BIS-11.
This section would benefit from a distinct procedure sub-section. Moreover, there is no statement of the predictor and outcome variables. The design needs to be explicitly stated (i.e., cross-sectional questionnaire). Moreover, as ANOVA is also reported in the statistical analyses section, outlining the three separate groups in a design sub-section would be ideal.
Could the authors map the hypotheses to the statistical analyses sub-section – i.e., To test hypothesis 1 XXX
Results
Lines 201-204 – what is meant by “higher than other groups” here? Do you mean MDD only and controls?
Overall, the results are reliant on tables with limited write up. The regression should have more detailed reporting, outlining the significance of the model, the proportion of variance explained, the findings of the Hosmer and Lemeshow test (for model fit), and the unit increase and influence on the OR.
There are far too many tables (6) for this section and very little results writing. It should not be left to the reader to have to spend a significant proportion of the time looking at the tables to unpick and interpret the key findings.
Discussion
The first paragraph should summarise the key predictions outlined at the end of the introduction, summarise key findings and how they support (or refute) the experimental hypothesis.
“Our study revealed that participants with MDD engaged in binge eating behaviors more than healthy individuals.” Isn’t this obvious given
Lines 331-333 indicate to me that the present study does not offer anything novel – it is common sense for those with depressive symptoms and binge eating symptoms to be more severe in those with MDD and BED compared to those without BED.
The authors state “Our study showed that binge eating behavior was higher and psychological pain was more severe in those with MDD and BED compared to healthy controls.” The interesting finding here would surely be whether psychological pain is more severe in those with MDD and BED compared to those with MDD? It is unclear why the comparison is only between individuals with both MDD and BED and HCs.
I would strongly recommend that the authors repeat the ANOVA with three groups (MDD+BED, MDD, HC). This would help determine whether BED adds an additional psychological burden above and beyond MDD. Without this comparison, it is hard to know whether BED exacerbates psychological pain beyond what is already present in depression.
The authors state “Although individuals turn to binge eating behavior to cope with negative emotions, it is controversial whether negative emotions decrease as a result of this behavior.” I feel this statement can be strengthened – for example “research is mixed on whether this behaviour actually reduces emotional distress – with some evidence suggesting only temporary relief followed by increased negative affect”.
The conclusion overall is fairly weak and not compelling. Again, the analyses reference ‘groups’ indicating more than two. But the entire discussion is focused on those with MDD+BED and controls, rather than those also without comorbid BED.
Author Response
Comment 1
Abstract — Line 25 – “Additionally, a negative correlation was observed between TMPS and the scores of 25 other scales.” Can the authors please clarify whether this means all other scales? It may be helpful to make this more explicit for the reader.
Response 1
We thank the dear reviewer for this constructive comment. We tried to explain the related sentences clearly. Please find the related sentence in the abstract below.
Additionally, a negative correlation was observed between TMPS and the scores of EDE-Q-13 bingeing, HDRS, BIS, and PS.
Comment 2
Abstract — Line 26 – “BED is a common comorbid eating disorder with MDD” this phrase can be deleted for conciseness.
Response 2
To address this comment, we deleted the related sentence from the abstract.
Comment 3:
Abstract — Line 29 – “The fact that”. Please could the authors maintain scientific accuracy and tone here. I would advise this is revised to “Our results suggest” or “The finding that”…
Response 3
To address this comment, we changed the statement ‘the fact that’ to ‘the finding that'. Please find the current version of the related sentence below:
The finding that binge eating behavior was most strongly associated with depression severity and body mass index (BMI) supports that binge eating behavior is a maladaptive attitude. Longitudinal studies comparing individuals with different BMIs in different clinical samples are needed to confirm our results.
Comment 4:
Keywords – It would be beneficial to include ‘major depressive disorder’ here.
Response 4
To address this comment, we revised the keywords. Please see the current version below.
Keywords: binge eating, depression, major depressive disorder, psychological pain, impulsivity
Comment 5:
Introduction – Spelling error – line 27 “accptd”
Response 5
We apologize for the typo. We corrected the related sentence.
“…it is accepted to develop due to the interaction of environmental and biological risk factors.”
Comment 6:
Lines 60-61: “Individuals with BED have an eating pattern in which they experience a subjective loss of control over their eating behaviors, and many are overweight, as compensatory behaviors do not accompany the picture.” – please can the authors provide an example of compensatory behaviours in parentheses. Additionally, ‘do not accompany the picture’ is rather informal. I’d suggest this is revised accordingly.
Response 6
We thank the dear reviewer for this constructive comment. We tried to improve the related sentence. Please find the current version below.
Individuals with BED display eating patterns characterized by a subjective loss of control over their eating behaviors. BED is often associated with increased reward sensitivity and impulsivity, which may lead to compulsive-like eating behavior (Chiappini et al., 2023). Many individuals affected by this disorder are overweight, as compensatory behaviors such as physical exercises and the use of laxatives do not occur.
Comment 7:
The description offered with respect to psychological pain requires more detail. Firstly, the authors should attempt to provide a definition based on previous research. For example, Meerwijk et al. (2013) describe psychological pain as a lasting, unsustainable and unpleasant feeling resulting from negative appraisal of an inability to protect the self from shame or harm, an ability to achieve something that is considered of vital importance, or a deficiency in love, support or affiliation. There is evidence to suggest that the posterior cingulate cortex plays a more prominent role in psychological (vs. physical) pain. See the below which may be helpful:
Response 7
To address this comment, the description of psychological pain has been expanded, and the explanations of the neurobiological underpinnings of this concept have been broadened, particularly emphasizing the posterior cingulate cortex. Please find the current version below.
“According to Meerwijk et al., psychological pain is a persistent and distressing feeling that occurs when individuals negatively perceive their inability to protect themselves from shame or harm, fail to achieve something deemed vital, or experience a lack of love, support, or connection with others. Although there is increasing evidence supporting the idea that psychological pain represents an independent phenomenon with its neurobiological properties (altered activities in brain regions such as the thalamus, anterior and posterior cingulate cortex, the prefrontal cortex, cerebellum, and parahippocampal gyrus)...”
Comment 8
I would recommend a paragraph being added which highlights the potential links between binge eating behaviour, psychological pain and major depressive disorder. There is evidence in this area. While ‘psychological pain’ is not always explicitly measured in such studies, this can be indirectly indicated via the intense emotions which encapsulate psychological pain:
(Tomba 2021, Ciwoniuk 2022, Duarte 2017)
Response 8
To address this comment, a new paragraph addressing potential associations between binge eating behavior, psychological pain and major depressive disorder was added to the introduction.
Eating disorders are not only characterized by impaired eating-related behaviors. Difficulties in tolerating negative emotional states, emotional dysregulation, and major depressive disorder often co-occur with eating disorders (Tomba et al., 2021). Women with anorexia nervosa experience anguish and frustration due to dissatisfaction with their appearance, thus experiencing more intense depressive symptoms and psychological pain (Ciwoniuk et al., 2023). Binge eating disorder is characterized by feelings of self-loathing, depression, or guilt after eating (APA 2013). Duarte et al. (2017) reported that emotional states such as shame and distress are predictors of binge eating.
We also have added the related references.
Ciwoniuk, N., Wayda-Zalewska, M., & Kucharska, K. (2023). Distorted body image and mental pain in anorexia nervosa. International Journal of Environmental Research and Public Health, 20, 718.
Duarte, C., Pinto‐Gouveia, J., & Ferreira, C. (2017). Ashamed and fused with body image and eating: Binge eating as an avoidance strategy. Clinical Psychology & Psychotherapy, 24(1), 195–202.
Tomba, E., Tecuta, L., Gardini, V., & Lo Dato, E. (2021). Mental pain in eating disorders: An exploratory controlled study. Journal of Clinical Medicine, 10(16), 3584.
Comment 9
The authors have not included any predictions based on previous research – based on the literature available what do the authors speculate may be the relationship between psychological pain and binge eating behaviours in individuals with MDD?
Response 9
To address this comment, the hypotheses at the end of the introduction were revised, and our expectations based on the existing literature were clearly stated. Please find the related sentences below.
“Based on the hypothesis that binge eating behaviors may be a dysfunctional, emotion-focused coping strategy, we aimed to examine the relationship between psychological pain and binge eating behaviors in individuals with MDD. We hypothesized that binge eating behavior would be exhibited more frequently and intensely in patients with MDD who experienced more intense psychological pain. Our second hypothesis was that depression severity and binge eating behavior would be correlated.”
Comment 10
Please change the term “mental retardation” to “intellectual disability” or similar. This is an outdated term.
Response 10
To address this comment, we changed the related statement. Please find the current version of the sentence below.
“In the MDD group, those diagnosed with intellectual disability, neurocognitive disorders, comorbid mental disorders, eating disorders other than binge eating disorder…”
Comment 11
What design was factored into the G power analysis? I presume a between-subjects design.
Response 11
We performed a priori power analysis to compare scale scores between the MDD and control groups. With a medium effect size (Cohen's d = 0.50), a power of 0.95, and a margin of error of 0.05 (p = 0.05), the minimum sample size required in a single group was calculated as 105 for the patient and control groups, totaling 210. With 275 participants, we assumed that the sample had sufficient power.
For your records, the details of the power analysis are:
Means: Difference between independent means
Analysis: A priori: Compute required sample size
Input: Tail(s) = Two
Effect size d = 0.5
α err prob = 0.05
Power (1-β err prob) = 0.95
Allocation ratio N2/N1 = 1
Output: Noncentrality parameter δ = 3.6228442
Critical t = 1.9714347 Df = 208
Sample size for the MDD group = 105
Sample size for the control group 2 = 105
Total sample size = 210
Actual power = 0.9501287
Comment 12
Can you please provide an example item from the HDRS, PS, TMPS-10, EDE-Q-13 and BIS-11.
Response 12
To address this comment, we tried to provide examples for the scales. Please find some of the examples below.
The Psychache Scale (PS): The PS is a 13-question self-report scale that assesses the severity of psychological pain. The first item of the PS is, “I feel psychological pain”.
The Tolerance for Mental Pain Scale-10 (TMPS-10): The TMPS-10 assesses the ability to tolerate psychological pain. The first item of the TMPS is 'I believe that my pain will go away'.
Comment 13
This section would benefit from a distinct procedure sub-section. Moreover, there is no statement of the predictor and outcome variables. The design needs to be explicitly stated (i.e., cross-sectional questionnaire). Moreover, as ANOVA is also reported in the statistical analyses section, outlining the three separate groups in a design sub-section would be ideal.
Response 13
To address this comment, a separate sub-section was created under the title “Procedure,” and the study design was clearly defined as a “cross-sectional questionnaire.”
2.2.1 Procedure
This study was conducted with a cross-sectional questionnaire design. Data were collected in three groups: (1) individuals with MDD and BED (MDD+BED+), (2) individuals with MDD without BED (MDD- only), and (3) healthy controls.
Descriptive statistics were summarized as mean ± standard deviation for continuous variables and number and percentage for categorical variables. The normality of the distribution of continuous variables was assessed using skewness and kurtosis values (acceptable range: -1.5 to +1.5) (Tabachnick and Fidell 2013). The chi-square test was used for categorical variable comparisons between groups.
To test the hypothesis that binge eating behavior will be more frequent and intense in individuals with MDD who experience more intense psychological pain (hypothesis 1), EDE-Q-13 bingeing subscale, PS, and TMPS-10 scores were compared between the three groups with one-way ANOVA. Homogeneity of variance was evaluated with Levene's test; the Tukey test was used when homogeneity was achieved, and the Tamhane test was used for post hoc analysis when homogeneity was not achieved.
While assessing the hypothesis that depression severity and binge eating behavior (hypothesis 2) would be correlated, Pearson correlation analysis was performed between the scores of the EDE-Q-13 bingeing subscale and HDRS, PS, BIS-11, and TMPS-10 scores in the MDD group.
We performed linear regression analysis to determine the factors predicting binge eating behavior in the MDD group (with and without BED). The dependent variable was the EDE-Q-13 bingeing subscale score. The independent variables were HDRS, PS, TMPS-10, and BMI. Variance Inflation Factor (VIF) values were examined for multicollinearity assessment; a VIF value below five was considered appropriate.
The statistical significance level was accepted as p < 0.05 in all analyses. Statistical analyses were performed using IBM SPSS 22.0 software (IBM Corporation, NY, USA).
Comment 14
Could the authors map the hypotheses to the statistical analyses sub-section – i.e., To test hypothesis 1 XXX
Response 14
To address this comment, we tried to map the hypotheses. Please find the related sentences below.
To test the hypothesis that binge eating behavior will be more frequent and intense in individuals with MDD who experience more intense psychological pain (hypothesis 1), EDE-Q-13 bingeing subscale, PS, and TMPS-10 scores were compared between the three groups with one-way ANOVA. Homogeneity of variance was evaluated with Levene's test; the Tukey test was used when homogeneity was achieved, and the Tamhane test was used for post hoc analysis when homogeneity was not achieved.
While assessing the hypothesis that depression severity and binge eating behavior (hypothesis 2) would be correlated, Pearson correlation analysis was performed between the scores of the EDE-Q-13 bingeing subscale and HDRS, PS, BIS-11, and TMPS-10 scores in the MDD group.
Comment 15
The participant sample numbers can be clearer to ease cognitive load on the reader. It would be best to state what the original sample was, then mention the participants excluded and that the “final sample consisted of XXX”.
Response 15
To address this comment, we tried to explain the sample in more detail. Please see the current version of the related sentence below.
In the MDD group, those diagnosed with intellectual disability, neurocognitive disorders, comorbid mental disorders, eating disorders other than binge eating disorder, and those diagnosed with chronic physical diseases such as malignancy and endocrine diseases were excluded by examining hospital files, self-statements of the participants and health system records because of their confounding effects on eating attitudes. After psychiatric interviews based on DSM-5 criteria, 10 individuals with comorbid anxiety disorders and 3 with alcohol use disorder were excluded. In addition, those diagnosed with atypical depression, one of the determinants of MDD, were also not included due to the presence of increased appetite among the diagnostic criteria. In the control group, participants with a diagnosis of mental disorder, psychotropic drug users, and those with a diagnosis of chronic physical illness were also excluded. After psychiatric interviews based on DSM-5 criteria, 2 participants with panic disorder and one with OCD were excluded. The final sample consisted of 147 participants with MDD and 128 healthy volunteers.
Comment 16
What is meant by “higher than other groups” here? Do you mean MDD only and controls?
Response 16
We apologize for the confusion in defining groups in the previous version of our article. We thank the dear reviewer for this constructive comment. We categorized the sample into three groups: MDD+BED+, MDD-only and the control group. We have revised the whole text and renamed the groups. Please find current version of the related sentence below:
Table 3 compares the PS, EDE-Q-13 total score, EDE-Q-13 bingeing subscale, TMPS-10, and HDRS scores of the participants (three groups). The EDE-Q-13 total and EDE-Q-13 bingeing scores of the MDD+BED+ group were significantly higher than the MDD-only (MDD+BED-) and the control groups (p<0.05 for each comparison).
Comment 17
Overall, the results are reliant on tables with limited write up. The regression should have more detailed reporting, outlining the significance of the model, the proportion of variance explained, the findings of the Hosmer and Lemeshow test (for model fit), and the unit increase and influence on the OR.
Response 17
Our regression model was statistically significant (F(4,142)=11.407, p<.001). The model explained 24.3% of the variance in the EDQ-Binge Eating score (R² = .243; Adjusted R² = .222). HDD and BMI variables were found to be significant predictors in the model (p<.05). No multicollinearity problem was observed (VIF<2.5). The analysis was performed using linear regression; therefore, logistic regression-specific outputs such as the Hosmer and Lemeshow test and the odds ratio are not included.
Please find the related paragraphs in the results:
Table 6 presents the linear regression model of the factors predicting binge eating behavior. Our regression model was statistically significant (F(4,142)=11.407, p<0.001). The model explained 24.3% of the variance in the EDE-Q-13 bingeing score (R² =0 .243; Adjusted R² =0 .222). The HDRS total score and BMI were the strongest predictors of EDE-Q-13 bingeing score (the dependent variable; p=0.011, p<0.001), respectively). PS and TMPS-10 scores did not have a significant predictive role in the model[zn1] . No multicollinearity problem was observed (VIF<2.5).
According to Cohen’s f² criteria, this corresponds to a medium-to-large effect size (f² = 0.32).
Comment 18:
There are far too many tables (6) for this section and very little results writing. It should not be left to the reader to have to spend a significant proportion of the time looking at the tables to unpick and interpret the key findings.
To address this comment, the findings were presented in the text in a more detailed and descriptive manner. Please find some of the new sentences below:
Response 18:
The participants with MDD were divided into two groups: those with and without comorbid BED (82 and 65 participants). Table 1 compares the sociodemographic characteristics of the groups. There was no statistically significant difference between the groups regarding sex, marital status, employment status, place of residence, income level, and family type (p>0.05 for each). These findings reveal that the groups are comparable in terms of demographic characteristics.
There was a statistically significant association between the groups and the BMI category, χ² (6, N = 274) = 33.48, p < 0.001. Post-hoc analysis revealed that individuals in the MDD+BED+ group were significantly more likely to be obese compared to MDD-only and control groups. In contrast, participants in the control group were significantly more likely to have a normal BMI.
Table 1. Sociodemographic characteristics of the participants
In Table 2, comparisons were made between two groups: 147 participants with MDD and 128 healthy controls. There were statistically significant differences between the groups for EDE-Q-13 bingeing, HDRS, TMPS-10, PS, and BIS-11 subscales and total scores and BMI (p<0.05 for each).
In addition to assessing statistical significance, effect sizes were calculated for all comparisons. The difference in EDE-Q-13 bingeing scores between the MDD group and the control group resulted in a large effect size (Cohen’s d = 1.04). The HDRS and the PS scores demonstrated very large effect sizes (d = 1.91 and d = 1.22, respectively). Medium to large effect sizes were observed in impulsivity scores, including the BIS attention (d = 0.58) and BIS non-planning (d = 0.62). Additionally, the TMPS-10 score revealed a large negative effect (d = -1.23), indicating that control participants reported higher scores than those in the MDD group.
Comment 19
The first paragraph should summarise the key predictions outlined at the end of the introduction, summarise key findings and how they support (or refute) the experimental hypothesis.
Response 19
To address this comment, we revised the discussion's first paragraph and tried to respond to the suggestions. Please find the related paragraph below.
MDD is a common psychiatric disorder that causes impairment, reduces productivity, and threatens public health. The psychological pain that often accompanies MDD affects the prognosis negatively. Consistent with our hypotheses, we determined that psychological pain and depressive symptoms are more severe in individuals with MDD when the diagnosis of BED is added and that individuals with BED tolerate psychological pain worse. Masheb and Grilo (2006) evaluated the frequency of emotional eating associated with feelings of anxiety, sadness, loneliness, fatigue, anger, and happiness in overweight individuals (BMI > = 25) diagnosed with BED. They demonstrated a significant rela-tionship between eating disorder symptoms and the severity of depressive symptoms (Masheb and Grilo 2006). The risk of being diagnosed with BED is approximately 5 times higher in those with MDD than in those without MDD (Öksüz and Orhan 2012). Masheb and Grilo (2006) did not determine the comorbid psychiatric disorders of the participants. Our findings indicated that binge eating behavior was significantly more pronounced in the MDD+BED+ and MDD-only groups compared to healthy controls, aligning with previous research. Furthermore, our results showed that the MDD+BED+ group experienced more severe depressive symptoms than the MDD-only group. This confirms that individuals with depression are more susceptible to emotional eating and are at a higher risk for developing eating disorders. These results support the existence of a bidirectional relationship between binge eating behaviors and depressive symptoms (Puccio et al., 2017).
Comment 20:
Indicate to me that the present study does not offer anything novel – it is common sense for those with depressive symptoms and binge eating symptoms to be more severe in those with MDD and BED compared to those without BED.
Response 20
To address this comment, we tried to present more clearly the possible contributions of our study to the existing literature. Please find the related paragraphs of the discussion below.
For individuals with BED, weight gain is a common outcome, as they typically do not engage in compensatory behaviors (Hudson et al., 2017). Research indicates that BED is strongly linked to obesity, with individuals affected by BED often having a higher BMI compared to the general population (Hudson et al., 2017; Grilo et al., 2021). It has also been reported that individuals with a high BMI exhibit increased reward sensitivity, making them more likely to pursue food for its rewarding properties (Dawe and Loxton, 2004; Meule and Platte, 2023). This supports our finding that BMI is one of the strongest predictors of BED symptoms. In contrast, eating behavior tends to be more regular and is regulated by homeostatic mechanisms in individuals without BED (Kessler et al., 2013). This explains why individuals in the control group had more normal BMI values compared to those in the MDD group. Moreover, negative affect is a precursor to disordered eating behaviors and an important vulnerability factor in the development of eating disorders. Binge eating is associated with both inadequate emotion regulation and expectations that eating will alleviate negative emotions. The relief of negative affect that increases before the eating behavior after the binge behavior may lead to binge eating serving as a method of coping with negative affect and reinforcing this behavior (Berg et al.2015; Wonderlich et al., 2021).
Our study presented a transdiagnostic approach to MDD and BED comorbidity by investigating the relationship between psychological pain and impulsivity. In addition, we evaluated the relationship between BMI, depressive symptoms, psychological pain, and psychological pain tolerance on BED with a multifactorial approach. The clinical contributions of our study include demonstrating that eating disorders should be routinely questioned in the evaluation of patients with MDD and the severity of psychological pain should be taken into account in clinical follow-up. Our study also suggests that approaches to improve emotion regulation skills in the MDD and BED comorbidity may be useful.
Comment 21
The authors state “Our study showed that binge eating behavior was higher and psychological pain was more severe in those with MDD and BED compared to healthy controls.” The interesting finding here would surely be whether psychological pain is more severe in those with MDD and BED compared to those with MDD? It is unclear why the comparison is only between individuals with both MDD and BED and HCs.
Response 21
To address this comment, we added a new paragraph to discuss the differences in psychological pain scores of the MDD+BED+ and MDD-only groups. Please see the related sentences below.
Similar to Tomba et al. (2021), binge eating behavior, depressive symptom severity, and psychological pain were correlated in patients with MDD in our study. However, when patients with MDD were classified according to whether BED accompanied them, the severity of psychological pain was similar, while the MDD+ BED+ group had more severe depressive symptoms. Our results suggest that the relationship between negative affect and binge eating is not specific to individuals with eating disorders at the clinical level, as emphasized by Berg et al. (2015). Therefore, questioning binge eating behavior in the treatment process of patients with MDD, even if an eating disorder does not accompany them, may help to identify maladaptive behaviors.
Comment 22
I would strongly recommend that the authors repeat the ANOVA with three groups (MDD+BED, MDD, HC). This would help determine whether BED adds an additional psychological burden above and beyond MDD. Without this comparison, it is hard to know whether BED exacerbates psychological pain beyond what is already present in depression.
Response 22
We appreciate the dear reviewer for the opportunity to revise our article regarding this matter. In the section on statistical analysis, we classified the sample into three groups: MDD+BED+, MDD-only, and the control group, and conducted statistical analyses accordingly. Below, you can find some tables comparing three groups.
Table 3. Comparison of Scale Scores Among Participants
|
Variables
|
Group |
Mean ± (sd) |
Significant |
Posthoc |
|
PS |
1MDD+BED + |
40.1 (12.9) |
<0.001 |
1>3* 2>3* 1>2 |
|
2MDD-only |
35.3 (13.7) |
|||
|
3Control |
24.1 (8.1) |
|||
|
EDE-Q-13-Total |
1MDD+BED + |
2.8 (0.9) |
<0.001 |
1>3# 1>2# 3>2 |
|
2MDD-only |
1.3 (0.8) |
|||
|
3Control |
1.4 (0.8) |
|||
|
EDE-Q-13-Bingeing |
1MDD+BED + |
3.3 (1.7) |
<0.001 |
1>3* 1>2* 2>3 |
|
2MDD-only |
0.7 (1.1) |
|||
|
3Control |
0.6 (0.8) |
|||
|
TMPS-10 |
1MDD+BED + |
31.2 (8.6) |
<0.001 |
3>1* |
|
2MDD-only |
33.2 (8.9) |
2>1 |
||
|
3Control |
41.9 (6.8) |
3>2* |
||
|
HDRS |
1MDD+BED + |
28.1 (7.8) |
<0.001 |
1>3* 1>2* 2>3* |
|
2MDD-only |
24.1 (8) |
|||
|
3Control |
4.4 (2.8) |
Note: PS: Psychache scale, EDE-Q-13: Eating Disorder Examination Questionnaire, HDRS: Hamilton Depression Rating Scale, TMPS: Tolerance for Mental Pain Scale, sd: standart deviation, MDD: Major Depressive Disorder, BED: Binge Eating Disorder
#: Tukey, *: Tamhane
Statistically significant differences are highlighted in bold.
Table 4. Comparison of Impulsivity Dimensions Among Participants
|
Variables |
Group |
Mean+ sd |
Significant |
Posthoc |
|
BIS-attention impulsivity |
1MDD+BED + |
17.6 (3.8) |
<0.001 |
1>3# 1>2 2>3# |
|
2MDD-only |
16.9 (4.3) |
|||
|
3Control |
15.1 (3.5) |
|||
|
BIS-motor impulsivity |
1MDD+BED + |
20.3 (4.8) |
0.073 |
1>3 1>2 2>3 |
|
2MDD-only |
19.4 (4.5) |
|||
|
3Control |
18.7 (4.0) |
|||
|
BIS-nonplanning impulsivity |
1MDD+BED + |
27.1 (5.0) |
<0.001 |
1>3* 1>2 2>3* |
|
2MDD-only |
25.6 (4.6) |
|||
|
3Control |
23.6 (4.5) |
|||
|
BIS-Total |
1MDD+BED + |
65.1 (10.4) |
<0.001 |
1>3* |
|
2MDD-only |
62.0 (10.6) |
1>2 |
||
|
3Control |
57.5 (9.7) |
2>3* |
Note: BIS: Barratt Impulsiveness Scale, sd: standart deviation, MDD: Major Depressive Disorder, BED: Binge Eating Disorder
#: Tukey, *: Tamhane
Statistically significant differences are highlighted in bold.
Comment 23
The authors state “Although individuals turn to binge eating behavior to cope with negative emotions, it is controversial whether negative emotions decrease as a result of this behavior.” I feel this statement can be strengthened – for example “research is mixed on whether this behaviour actually reduces emotional distress – with some evidence suggesting only temporary relief followed by increased negative affect”.
Response 23
To address this comment, we tried to explain the related statements clearly and in more detail.
Although individuals turn to binge eating behavior to cope with negative emotions, the findings are controversial on whether this behavior actually reduces emotional distress – with some evidence suggesting only temporary relief followed by increased negative affect. Stein et al. (2007) evaluated 33 overweight women with BED according to DSM-IV criteria and examined possible associations between binge eating behavior and pleasantness, mood, guilt, and arousal states while the participants were binge eating or eating regularly.
Comment 24:
“The conclusion overall is fairly weak and not compelling.”
“Again, the analyses reference ‘groups’ indicating more than two. But the entire discussion is focused on those with MDD+BED and controls, rather than those also without comorbid BED.”
Response 24
To address this comment, we revised the conclusion. Please find the current version below.
Conclusions
Our results demonstrate that MDD can be characterized by a wide range of symptoms, such as severe psychological pain, impulsivity, and binge eating behavior, in addition to typical depressive symptoms. The difference in depressive symptom severity in patients with MDD+BED+ compared to the MDD-only group suggests that negative affect would play a leading role and that individuals resort to binge eating for affect regulation. The finding that binge eating behavior is not associated with the trait impulsivity may be interpreted as a maladaptive coping strategy instead of losing control over eating. Especially in patients with MDD with high BMI, binge eating behavior should be questioned, and adaptive affect regulation methods should be targeted in treatment. Future studies could examine these findings using different samples and longitudinal designs.

Reviewer 3 Report
Comments and Suggestions for Authors
This cross-sectional study analysed the relationship between psychological pain and binge eating behaviours in individuals with major depressive disorder. 275 participants were included in this study and were assessed for their depression, psychological pain, tolerance for mental pain, impulsiveness, and eating disorder pathology. The authors found correlations between EDE-Q bingeing, HDRS, BIS, PS and TMPS.
Major comments:
- Methods:
- Include the measurements of weight and height: based on anamnesis or actual measurement at the research centre? Explain when it was measured and how it was measured.
- Include the reporting guidelines and provide the checklist in the supplement, e.g. STROBE
- Please explain some of the terms further: salary cut-off, family type, and smaller than provincial centre.
- I would suggest clarifying and using this term clearly: gender or sex?
- How would the authors classify marital status for people who were divorced or cohabiting together?
- I would suggest including ethnicity, if possible.
- Results:
- Include effect sizes
- I would suggest including BMI categories in the demographic table (normal BMI, overweight, obese) to expand the points for discussion.
- If the authors have the data, I would suggest including the length of MDD and BED in the table.
- Discussion:
- “Our study revealed that participants with MDD engaged in binge eating behaviors more than healthy individuals.”: I would suggest not including this, as it could be misleading from this study. The authors only included healthy participants (without BED) as the control group. Thus, it is not a comparable comparison.
- “In addition, we found that both depressive symptoms and binge eating symptoms were more severe in those in the group of MDD with BED than those without BED.”: I suggest removing this sentence for the same reason as above.
- “We also investigated the factors associated with binge eating behavior and found that the strongest predictors were depressive symptom severity and BMI.” I assume this is based on patients with MDD? If so, please clarify the sentence.
- After grouping the BMI of the participants in this study (normal, overweight, obese), I would suggest that the authors elaborate the findings with wider literature
- I suggest exploring further why BIS, TMPS, and PS were nonsignificant between MDD+BED+ and MDD+BED-. For example, as MDD, alongside BED, is not included as an impulsive disorder, why did the PIS results show higher scores compared to normal?
- Please include the strengths and clinical implications of this study
Minor comments:
- Abstract: include the effect sizes alongside the p-values
- The introduction is well written and easy to follow.
- Typing error in line 47
- Include updated studies, e.g. https://doi.org/10.1002/eat.23648, https://doi.org/10.1002/eat.22401, https://doi.org/10.1002/eat.22410
- Methods:
- Include the month, year, and duration of the study
- Include HDRS abbreviations during the first mention
- Include education background of the interviewer
- Results:
- I suggest to revise the tables:
- Include the total number of participants in the table columns, e.g. MDD+/BED+ (n=82)
- Include n(%) in row title, e.g. gender n(%), marital status n(%)
- BMI kg/m2
- Use dot (.) for decimal values
- Use <0.001 instead of 0.000
- Please clarify the importance of Table 2. If it does not provide additional information for discussion, I suggest removing it or including it in the supplement.
- Discussion:
- grammatical error in line 344
- redundant information in lines 348-355. I would suggest removing the repeated sentences.
Author Response
Major Comments
Comment 1
Include the measurements of weight and height: based on anamnesis or actual measurement at the research centre?
Response
To address this comment, we clarified that the first author measured the height and weight directly after the psychiatric interview. Please find the related sentence below.
“The first author made height and weight measurements after the psychiatric interview. The body mass index (BMI) was calculated by dividing weight in kilograms by the square of height in meters (kg/m²).”
Comment 2
Include the reporting guidelines and provide the checklist in the supplement, e.g. STROBE.
Response
To address this comment, we added a supplementary file to prove that our manuscript adheres to the STROBE guidelines. Please find the STROBE table below.
STROBE Checklist
|
Item |
STROBE Recommendation |
Assessment/Explanation |
Reported Page(s) |
|
1a |
Study design indicated in title or abstract |
Not stated in title or abstract |
Page 1 |
|
1b |
Informative and balanced summary in abstract |
Objectives, methods, results, and conclusion are summarized |
Page 1 |
|
2 |
Scientific background and rationale |
Psychological pain, impulsivity, and BED relationship explained with references |
Pages 2–4 |
|
3 |
Objectives and hypotheses |
Two hypotheses clearly stated |
Page 4 |
|
4 |
Study design stated early |
'Cross-sectional questionnaire design' mentioned |
Page 8 |
|
5 |
Setting and data collection dates |
Balcalı Hospital, CU Psychiatry outpatient clinic, 01.10.2023 – 15.01.2024 |
Page 4 |
|
6a |
Eligibility criteria and participant selection |
Inclusion and exclusion criteria detailed |
Pages 4–5 |
|
6b |
Matching criteria for matched studies |
Not applicable |
— |
|
7 |
Definition of variables |
All scales and definitions (e.g., BMI) provided |
Pages 5–6 |
|
8 |
Sources and measurement methods |
Clinical interviews and validated scales described |
Pages 5–7 |
|
9 |
Bias minimization |
Comorbid exclusion, power analysis, validated tools |
Page 5 |
|
10 |
Study size estimation |
Explained using G*Power |
Page 5 |
|
11 |
Handling of quantitative variables |
Continuous variables like HDRS, BMI explained |
Pages 6–7 |
|
12a |
Statistical methods |
ANOVA, regression, correlation described |
Pages 7–8 |
|
12b |
Subgroup/interactions |
MDD+BED+, MDD only, and control group compared |
Pages 7–8 |
|
12c |
Missing data handling |
Not reported |
— |
|
12d |
Sampling strategy |
Not described |
— |
|
12e |
Sensitivity analyses |
Not conducted |
— |
|
13a |
Participants at each stage |
Final sample size and excluded cases described |
Page 5 |
|
13b |
Non-participation reasons |
DSM-based exclusions explained |
Page 5 |
|
13c |
Flow diagram |
Not included |
— |
|
14a |
Descriptive data |
Sociodemographics in Table 1 |
Page 9 |
|
14b |
Missing data per variable |
Not mentioned |
— |
|
15 |
Outcome data |
Scale scores (EDE-Q, HDRS, TMPS, etc.) reported |
Pages 9–12 |
|
16a |
Main results with estimates |
ANOVA, post-hoc, regression reported |
Pages 9–12 |
|
16b |
Category boundaries |
WHO BMI classification used |
Page 5 |
|
16c |
Other analyses |
Correlations and regressions conducted |
Pages 9–12 |
|
18 |
Summary of key results |
Linked back to hypotheses |
Page 12-13 |
|
19 |
Study limitations |
Cross-sectional nature, no gender analysis |
Page 15 |
|
20 |
Interpretation of results |
Interpreted with literature comparison |
Pages 13-16 |
|
21 |
Generalisability |
Limited to clinical sample |
Page 16 |
|
22 |
Funding/conflict of interest |
No funding, thesis-based, no conflicts |
Page 16 |
|
23 |
Ethics approval |
Ethics committee approval and Helsinki compliance |
Page 15 |
Comment 3:
Explain some of the terms further: salary cut-off, family type, and smaller than provincial centre.
Response
To address this comment, we tried to explain the related terms in more detail. Please find the related sentences below.
The participants' income levels were divided into below- and above-minimum wage based on the minimum wage determined by the Turkish government (approximately 550 USD per month). The family type was divided into two categories according to the household structure. When the household consists of parents and children, it was defined as a nuclear family; when it also consisted of other relatives, it was defined as an extended family. Place of residence were divided into two groups: 'provincial center' when they were city centers and 'settlement units smaller than the provincial center' when they were smaller in scale, such as districts, towns, or villages.
Comment 4:
Clarify and consistently use the term “gender” or “sex”.
Response
To address this comment, we revised the term across the manuscript and consistently used “sex,” as the variable was collected as biological sex. Please find the related sentence below.
Finally, although MDD is more common in women than men, the diagnosis of MDD was not evaluated according to sex in our study. Future studies can be designed to reveal the differences between the sexes.
Comment 5:
How would the authors classify marital status for divorced or cohabiting participants?
Response
To address this comment, we clarified that divorced participants were included in the “single” category. There were no participants living cohabiting in the sample.
Comment 6:
Include ethnicity, if possible.
Response
We state the reviewers and the editorial that ethnicity data were not collected in our study.
Comment 7
Include effect sizes.
Response
To address this comment, we added effect size statistics in the results section. Please find the related sentences below. We added the related sentences below each table in the manuscript.
There was a statistically significant association between the groups and the BMI category, χ² (6, N = 274) = 33.48, p < 0.001. Post-hoc analysis revealed that individuals in the MDD+BED+ group were significantly more likely to be obese compared to MDD-only and control groups. In contrast, participants in the control group were significantly more likely to have a normal BMI.
In addition to assessing statistical significance, effect sizes were calculated for all comparisons. The difference in EDE-Q-13 bingeing scores between the MDD group and the control group resulted in a large effect size (Cohen’s d = 1.04). The HDRS and the PS scores demonstrated very large effect sizes (d = 1.91 and d = 1.22, respectively). Medium to large effect sizes were observed in impulsivity scores, including the BIS attention (d = 0.58) and BIS non-planning (d = 0.62). Additionally, the TMPS-10 score revealed a large negative effect (d = -1.23), indicating that control participants reported higher scores than those in the MDD group.
ANOVA analyses revealed significant effect sizes. The difference in EDE-Q-13 total scores between the groups showed a large effect (η² = 0.36), indicating substantial differences among the three groups. Moderate to large effects were observed for the BIS non-planning and BIS total scores (η² = 0.10 for each). The effect size for BIS attention was moderate (η² = 0.08), while BIS motor impulsivity had a small effect (η² = 0.02
Pearson correlation coefficients, when interpreted as effect sizes, indicated moderate to strong associations among the variables. For example, binge eating showed a positive correlation with the HDRS at r = 0.398, with the PS at r = 0.273, and with the BIS at r = 0.233. Conversely, it displayed a negative correlation with the TMPS-10 at r = -0.257. The strongest association was observed between TMPS-10 and PS, which had a large negative effect size of r = -0.623.
According to Cohen’s f² criteria, this corresponds to a medium-to-large effect size (f² = 0.32). Among the predictors, both HDRS and BMI had significant and meaningful contributions to the model.
Comment 8:
Include BMI categories in demographic table.
Response:
To address this comment, we updated Table 1 to present BMI categories. Please find the current version of the releated table below.
Table 1. Sociodemographic Characteristics of the Participants
|
Variables |
MDD+BED+ (n=82)
|
MDD-only (n=65)
|
Control (n=128)
|
p |
|
Sex n(%) |
||||
|
Female |
60(73.2) |
42(64.6) |
91(71.1) |
0.50 |
|
Male |
22(26.8) |
23(35.4) |
37(28.9) |
|
|
Marital Status n(%) |
||||
|
Married |
41(50.0) |
34(52.3) |
54(42.2) |
0.32 |
|
Single |
41(50.0) |
31(47.7) |
74(57.8) |
|
|
BMI |
28.25±5.64 |
24.95±4.88 |
24.06±4.14 |
0.36 |
|
BMI Categorizations n(%) Underweight Normal weight Overweight Obese |
0(0) 25(30.5) 32(39) 25(30.5) |
5(7.8) 31(48.4) 19(29.7) 9(14.1) |
7(5.5) 77(60.2) 34(26.6) 10(7.8) |
p<0.001 |
|
Employment Status n(%) |
||||
|
Employed |
43(52.4) |
30(46.2) |
71(55.5) |
0.47 |
|
Unemployed |
39(47.6) |
35(53.8) |
57(44.5) |
|
|
Place of Residence n(%) |
||||
|
Provincial Center |
65(79.3) |
53(81.5) |
102(79.7) |
0.93 |
|
Smaller than Provincial Center |
17(20.7) |
12(18.5) |
26(20.3) |
|
|
Income Level n(%) |
||||
|
Below Monthly Minimum Wage |
27(32.9) |
17(26.2) |
46(35.9) |
0.39 |
|
Above Monthly Minimum Wage |
55(67.1) |
48(73.8) |
82(64.1) |
|
|
Family Type n(%) |
||||
|
Immediate Family |
71(86.6) |
54(83.1) |
117(91.4) |
0.21 |
|
Extended Family |
11(13.4) |
11(16.9) |
11(8.6) |
|
The wage refers to the legally mandated lowest salary that an employer is required to pay employees in the country.
BMI: Body Mass Index, MDD: Major Depressive Disorder, BED: Binge Eating Disorder
Comment 9
If the authors have the data, I would suggest including the length of MDD and BED in the table.
Response
We are sorry to inform you that we do not have the data about the length of MDD and BED.
Comment 10
“Our study revealed that participants with MDD engaged in binge eating behaviors more than healthy individuals.”: I would suggest not including this, as it could be misleading from this study. The authors only included healthy participants (without BED) as the control group. Thus, it is not a comparable comparison.
Response
To address this comment, we revised the language throughout the discussion section to eliminate potentially misleading comparative statements between individuals with MDD and healthy controls, specifically concerning binge eating behavior. Instead of stating that binge eating is more prevalent in individuals with MDD compared to healthy individuals, we rephrased the relevant sentences to clarify that the comparison refers to symptom scores rather than a diagnosis of BED. Please find the related sentence below.
Furthermore, our results showed that the MDD+BED+ group experienced more severe depressive symptoms than the MDD-only group. This confirms that individuals with depression are more susceptible to emotional eating and are at a higher risk for developing eating disorders.”
Binge eating behavior, depressive symptom severity, and psychological pain were correlated in patients with MDD in our study. However, when patients with MDD were classified according to whether BED accompanied them, the severity of psychological pain was similar, while the MDD+ BED+ group had more severe depressive symptoms.”
Comment 11
“In addition, we found that both depressive symptoms and binge eating symptoms were more severe in those in the group of MDD with BED than those without BED.”: I suggest removing this sentence for the same reason as above.
Response
To address this comment, we revised the relevant discussion sentence to avoid overstating group differences as diagnostic conclusions. Rather than stating symptom severity as an absolute or categorical comparison, we reframed the sentence to reflect group-level symptom scores within the MDD sample. Thus, the content was not removed but carefully revised to prevent potential misinterpretation. Please find the related sentence below.
“Furthermore, our results showed that the MDD+BED+ group experienced more severe depressive symptoms than the MDD-only group. This confirms that individuals with depression are more susceptible to emotional eating and are at a higher risk for developing eating disorders.”
Comment 12
“We also investigated the factors associated with binge eating behavior and found that the strongest predictors were depressive symptom severity and BMI.” I assume this is based on patients with MDD? If so, please clarify the sentence.
Response
To address this comment, we revised the sentence in the discussion section to explicitly state that the regression analysis was performed within the whole MDD group (n=147), including both individuals with and without BED. Please find the related sentence below.
Table 6 presents the linear regression model of the factors predicting binge eating behavior. Our regression model was statistically significant (F(4,142)=11.407, p<0.001). The model explained 24.3% of the variance in the EDE-Q-13 bingeing score (R² =0 .243; Adjusted R² =0 .222). The HDRS total score and BMI were the strongest predictors of EDE-Q-13 bingeing score (the dependent variable; p=0.011, p<0.001), respectively). PS and TMPS-10 scores did not have a significant predictive role in the model[zn1] . No multicollinearity problem was observed (VIF<2.5). According to Cohen’s f² criteria, this corresponds to a medium-to-large effect size (f² = 0.32).
Comment 13
After grouping the BMI of the participants in this study (normal, overweight, obese), I would suggest that the authors elaborate the findings with wider literatüre.
Response
To address this comment, we grouped BMI values using the World Health Organization classification system and integrated a broader literature framework to contextualize the relationship between BED, BMI, and reward sensitivity in the discussion. We also highlighted that BED is commonly comorbid with obesity, and that high BMI may be functionally linked to emotional eating and reward-driven behavior. Please find the related sentences below.
“The prevalence of obesity in the group with both MDD+BED+ was higher than in the MDD-only and control groups. For individuals with BED, weight gain is a common outcome, as they typically do not engage in compensatory behaviors (Hudson et al., 2017). Research indicates that BED is strongly linked to obesity, with individuals affected by BED often having a higher BMI compared to the general population (Hudson et al., 2017; Grilo et al., 2021). It has also been reported that individuals with a high BMI exhibit increased reward sensitivity, making them more likely to pursue food for its rewarding properties (Dawe and Loxton, 2004; Meule and Platte, 2023). This supports our finding that BMI is one of the strongest predictors of BED symptoms
Comment 14
I suggest exploring further why BIS, TMPS, and PS were nonsignificant between MDD+BED+ and MDD+BED-. For example, as MDD, alongside BED, is not included as an impulsive disorder, why did the BIS results show higher scores compared to normal?
Response
To address this comment, we expanded the discussion section to provide possible explanations for the non-significant differences observed in BIS, TMPS, and PS scores between MDD+BED+ and MDD-only groups. We emphasized the theoretical distinction between impulsivity as a state versus trait and discussed how general impulsivity levels may reflect the underlying characteristics of depression rather than being specific to BED comorbidity. Please find the related sentences below.
“Our results determine that the severity of impulsivity is not significantly different between those with MDD and BED together and those with MDD alone; this finding supports the hypothesis that BED is not in the same cluster with impulsivity and related disorders and that binge eating behavior serves the purpose of emotion regulation rather than impulsivity in the classification systems such as DSM-5 (Stein et al., 2007; APA 2013). In addition, the concepts of psychological pain and tolerance to psychological pain are closely related to MDD and negative affect. However, impulsivity is a more independent concept classified as a state or trait. In our study, the questioning of general impulsivity tendencies instead of state impulsivity measurement may have reflected the personality characteristics of participants with MDD independently of BED diagnosis (Antons and Brand 2018).”
Comment 15
Please include the strengths and clinical implications of this study
Response
To address this comment, we added a paragraph at the end of the discussion section highlighting the strengths of the study and outlining its clinical implications. Please find the related sentences below.
“Our study presented a transdiagnostic approach to MDD and BED comorbidity by investigating the relationship between psychological pain and impulsivity. In addition, we evaluated the relationship between BMI, depressive symptoms, psychological pain, and psychological pain tolerance on BED with a multifactorial approach. The clinical contributions of our study include demonstrating that eating disorders should be routinely questioned in the evaluation of patients with MDD and the severity of psychological pain should be taken into account in clinical follow-up. Our study also suggests that approaches to improve emotion regulation skills in the MDD and BED comorbidity may be useful.”
Comment 16:
Abstract: include the effect sizes alongside the p-values.
Response
To address this comment, we revised the abstract to include effect sizes for key comparisons alongside their corresponding p-values. Please find the related sentence below.
“In the group of MDD with BED comorbidity, EDE-Q-13 total, and bingeing subscale, HDRS scores were significantly higher than the other groups (p<0.05 for each group), with large to very large effect sizes (e.g., EDE-Q-13 bingeing d = 1.04; HDRS d = 1.91; PS d = 1.22).”
Minor Comments
Comment 1
The introduction is well written and easy to follow.
Response
We thank the dear reviewer for the valuable comment.
Comment 2
Typing error in line 47
Response
To address this comment, we corrected the typographical error at line 47 in the introduction section. Please find the related sentence below.
“…it is accepted to develop due to the interaction of environmental and biological risk factors.”
Comment 3
Include updated studies, e.g. https://doi.org/10.1002/eat.23648, https://doi.org/10.1002/eat.22401, https://doi.org/10.1002/eat.22410
Response
To address this comment, we integrated the suggested updated references into the introduction and discussion sections where relevant. Please find the related sentences below.
Moreover, negative affect is a precursor to disordered eating behaviors and an important vulnerability factor in the development of eating disorders. Binge eating is associated with both inadequate emotion regulation and expectations that eating will alleviate negative emotions. The relief of negative affect that increases before the eating behavior after the binge behavior may lead to binge eating serving as a method of coping with negative affect and reinforcing this behavior (Berg et al.2015; Wonderlich et al., 2021).
The relationship between the affect regulation model described above and binge eating has been well documented in the literature (Berg et al., 2015; Ambwani et al., 2015). Consistent with the affect regulation model, Berg et al. (2015) suggested that binge eating reduces negative affect and guilt, albeit short-term (Berg et al., 2015).
Comment 4
Include the month, year, and duration of the study.
Response
To address this comment, we tried to present the current study's details in the materials and methods. Please find the related sentence below.
This research adhered to the ethical principles of the Helsinki Declaration (2013) for human studies and was conducted between October 1, 2023, and January 15, 2024.
Comment 5
Include HDRS abbreviations during the first mention.
Response
To address this comment, we included the abbreviation HDRS when the Hamilton Depression Rating Scale was first introduced. Please find the related sentence below.
“We administered a sociodemographic data form, Hamilton Depression Rating Scale (HDRS), Psychache Scale (PS), Tolerance for Mental Pain Scale-10 (TMPS-10), Barratt Impulsiveness Scale (BIS-11), and Eating Disorder Examination Questionnaire (EDE-Q-13).”
Comment 6
Include education background of the interviewer.
Response
To address this comment, we clarified the educational background and training level of the interviewer in the “Measures” section. Please find the related sentence below.
“The primary author, a final-year student (MD) specializing in mental health and diseases, conducted interviews with all participants lasting approximately 40 to 60 minutes, under the supervision of the secondary author, an associate professor in psychiatry.”
Comment 7
I suggest to revise the tables: Include the total number of participants in the table columns, e.g. MDD+/BED+ (n=82); Include n(%) in row title, e.g. gender n(%), marital status n(%); BMI kg/m²; Use dot (.) for decimal values; Use <0.001 instead of 0.000; Please clarify the importance of Table 2. If it does not provide additional information for discussion, I suggest removing it or including it in the supplement.
Response
To address this comment, we revised all tables for clarity, consistency, and formatting. Specifically, we added total participant numbers in the column headers (e.g., MDD+BED+ (n=82)), included “n(%)” in all relevant row titles, ensured use of dot (.) for decimals, reported all p-values as "<0.001" where appropriate, and corrected units such as "BMI kg/m²". Regarding Table 2, we retained it as it summarizes key between-group comparisons that support the interpretation in the results and discussion. Please find the related sentences and table headings below.
Table 1: Column titles: “MDD+BED+ (n=82) | MDD only (n=65) | Control (n=128)”
Example row title with n(%): “Sex n(%)” Example decimal notation and p-value formatting from Table 2: “EDE-Q-13 Bingeing: 2.20 (1.97) vs. 0.60 (0.80), t = 8.541, p < 0.001”
Comment 8
Discussion: grammatical error in line 344; redundant information in lines 348–355. I would suggest removing the repeated sentences.
Response
To address this comment, we corrected the grammatical error noted in line 344 and removed redundant or repetitive content between lines 348–355 in the discussion section to improve clarity and avoid unnecessary duplication.

Round 2
Reviewer 1 Report
Comments and Suggestions for Authors
Paper can be published
Author Response
We thank the dear reviewer for the valuable comments about our manuscript.
Reviewer 2 Report
Comments and Suggestions for Authors
After reading the response from authors I am satisfied that my comments have been comprehensively addressed. I do believe the manuscript is now much stronger - and thus can recommend this for publication.
Author Response

(The authors gave the same response as above.)

Reviewer 3 Report
Comments and Suggestions for Authors
The authors have addressed all the comments.
To make the data in the Tables clearer, I suggest including the effect sizes in each table by creating a new column for effect sizes before p-values. For example, the columns in Table 1 would be better if they consisted of: variables, MDD+BED+, MDD-only, control, effect sizes, and p-value.
Author Response
To address this comment, we reorganised the related tables. Please see the current version of table 1 below.
Table 1. Sociodemographic Characteristics of the Participants
|
Variables |
MDD+BED+ (n=82)
|
MDD-only (n=65)
|
Control (n=128)
|
Effect size (Cramér’s) |
p |
|
Sex n(%) |
|
||||
|
Female |
60(73.2) |
42(64.6) |
91(71.1) |
|
0.50 |
|
Male |
22(26.8) |
23(35.4) |
37(28.9) |
||
|
Marital Status n(%) |
|
||||
|
Married |
41(50.0) |
34(52.3) |
54(42.2) |
|
0.32 |
|
Single |
41(50.0) |
31(47.7) |
74(57.8) |
||
|
BMI |
28.25±5.64 |
24.95±4.88 |
24.06±4.14 |
|
0.36 |
|
BMI Categorizations n(%) Underweight Normal weight Overweight Obese |
0(0) 25(30.5) 32(39) 25(30.5) |
5(7.8) 31(48.4) 19(29.7) 9(14.1) |
7(5.5) 77(60.2) 34(26.6) 10(7.8) |
0.247 |
p<0.001 |
|
Employment Status n(%) |
|
||||
|
Employed |
43(52.4) |
30(46.2) |
71(55.5) |
|
0.47 |
|
Unemployed |
39(47.6) |
35(53.8) |
57(44.5) |
||
|
Place of Residence n(%) |
|
||||
|
Provincial Center |
65(79.3) |
53(81.5) |
102(79.7) |
|
0.93 |
|
Smaller than Provincial Center |
17(20.7) |
12(18.5) |
26(20.3) |
||
|
Income Level n(%) |
|
||||
|
Below Monthly Minimum Wage |
27(32.9) |
17(26.2) |
46(35.9) |
|
0.39 |
|
Above Monthly Minimum Wage |
55(67.1) |
48(73.8) |
82(64.1) |
||
|
Family Type n(%) |
|
||||
|
Immediate Family |
71(86.6) |
54(83.1) |
117(91.4) |
|
0.21 |
|
Extended Family |
11(13.4) |
11(16.9) |
11(8.6) |
||
The wage refers to the legally mandated lowest salary that an employer is required to pay employees in the country.
BMI: Body Mass Index, MDD: Major Depressive Disorder, BED: Binge Eating Disorder